# AKORN: Adaptive Knots generated Online for RegressioN splines

**Sunil Madhow** [1]   **Dheeraj Baby** [2]   **Yu-Xiang Wang** [1]

## Abstract

In order to attain optimal rates, state-of-the-art algorithms for non-parametric regression require that a hyperparameter be tuned according to the smoothness of the ground truth (Tibshirani, 2014). This amounts to an assumption of oracle access to certain features of the data-generating process. We present a *parameter-free* algorithm for offline non-parametric regression over $TV_1$-bounded functions. By feeding offline data into an optimal online denoising algorithm styled after (Baby et al., 2021), we are able to use change-points to adaptively select knots that respect the geometry of the underlying ground truth. We call this procedure AKORN (Adaptive Knots generated Online for RegressioN splines). By combining forward and backward passes over the data, we obtain an estimator whose empirical performance is close to Trend Filtering (Kim et al., 2009; Tibshirani, 2014), even when we provide the latter with oracle knowledge of the ground truth's smoothness.

## 1. Introduction

When estimating a nonparametric function with noisy data, the key challenge is knowing where to smooth observations and by how much. Because the "wiggliness" of the ground truth is unknown, practitioners are almost always left with a hyperparameter to tune, which corresponds to the wiggliness of the fit. Attaining optimal statistical rates often requires this parameter to be tuned with oracle knowledge of the ground truth. In this paper, we propose a (near)-optimal, *parameter-free* algorithm for non-parametric regression that uses techniques from online learning to automatically adapt to the smoothness of the ground truth.

---

[1]Halicioglu Data Science Institute, UC San Diego [2]Amazon (Work was completed prior to joining.). Correspondence to: Sunil Madhow <smadhow@ucsd.edu>.

*Proceedings of the 42$^{nd}$ International Conference on Machine Learning*, Vancouver, Canada. PMLR 267, 2025. Copyright 2025 by the author(s).

Consider the problem of non-parametric regression over total variation smoothness classes. For some covariates $\{x_i\}_{i=1}^n$, we observe data

$$y_i = f(x_i) + \epsilon_i \tag{1}$$

where $\{\epsilon_i\}$ are i.i.d. $\mathcal{N}(0, \sigma^2)$ random variables and $f$ has bounded $k$-th order total variation, which means that the variation in its $k$th derivative is controlled.

Traditionally, the best solutions to this problem solve functional risk-minimization objectives with regularization on $TV_k$-smoothness (Tibshirani, 2014). In order to enjoy optimal statistical rates, such methods require the regularization to be tuned in correspondence with a tight upper-bound on the $TV_k$ of the ground truth. This makes it difficult for practitioners to be sure that they are benefiting from the powerful theory already established in the literature (Tibshirani, 2014; Guntuboyina et al., 2020).

Recent work uses algorithms from the Online Learning (OL) literature (Hazan et al., 2006; Baby et al., 2021; Chatterjee & Goswami, 2023) to treat the online version of this regression problem, where points $(x_i, y_i)$ are revealed one at a time. Thanks to the powerful oracle inequalities enjoyed by OL algorithms, these methods attain optimal[1] statistical rates while obviating the necessity for a priori knowledge of the smoothness of $f$.

When applied directly in the offline setting, however, existing OL-based methods have serious drawbacks. Principal among these is the fact that their output is not a function $\hat{f}$, but rather a highly non-smooth sequence of predictions, $\{\hat{y}_1, ... \hat{y}_n\}$. This is problematic because inferring a *smooth*, functional form is one of the key goals in the regression literature (Donoho & Johnstone, 1994). At the same time, each prediction, $\hat{y}_t$, is made with only knowledge of $y_1, ... y_{t-1}$, making it harder to pick up patterns in the data. The result is that, when specialized to the offline setting (for instance, by interpolating the predictions $\hat{y}_1, ... \hat{y}_n$), online algorithms are badly outperformed by traditional methods in terms of both MSE and attractiveness of fit (Baby et al., 2021).

Is it possible to inject the instance-dependent knowledge acquired by online algorithms into inherently *offline* algo-

---

[1]Optimal rates for online and offline regression over $TV$ classes differ only in lower-order terms

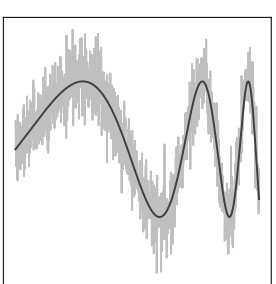 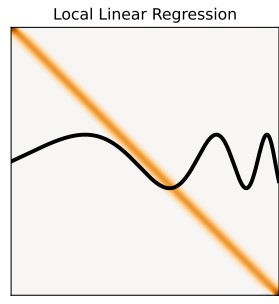 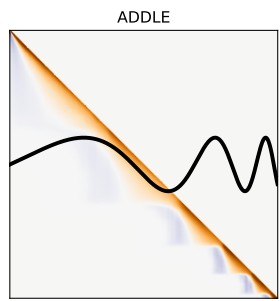 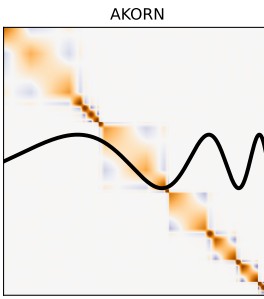

Local Linear Regression | ADDLE | AKORN

*Figure 1.* "Attention Map" for AKORN compared to ADDLE and Local Linear Regression for noisy evaluations of the Doppler function of (Donoho & Johnstone, 1994). Observe that ADDLE and AKORN can select the appropriate "bandwidth" for the local linear fit adaptively.

rithms? In this paper, we present an OL-based algorithm called AKORN (Adaptive Knots generated Online for RegressioN splines) for offline non-parametric regression that retains some of the best properties of online and offline methods:

1. AKORN adapts to the smoothness of the ground truth with no need for hyperparameter tuning. That is, without any knowledge of $TV_1[f]$, AKORN outputs a linear spline, $\hat{f}$, satisfying

$$\frac{1}{n}\sum_{i=1}^{n}(\hat{f}(x_i) - f(x_i))^2 = \tilde{O}_P(n^{-4/5}TV_1[f]^{2/5})$$

Furthermore, $\hat{f}$ is guaranteed to have a number of knots scaling as $\tilde{O}(n^{1/5}TV_1[f]^{2/5})$.

2. AKORN learns a function, $\hat{f}$, rather than a sequence of point predictions. As such, AKORN offers a principled way of using online methodology for inference, rather than pure prediction.

3. We can visualize AKORN's "attention map" as in Figure 1. This highlights AKORN's ability to optimize a bias-variance tradeoff in a neighborhood of each covariate $x_i$ – a property inherited from its online subroutines. Details on the attention map are in Section 5.1.

4. AKORN enjoys the optimal statistical rate for the offline regression problem without requiring us to restrict our attention to functions whose $TV_1$ is bounded by a known constant. That is, the rate in item 1 is an instance-dependent rate that holds over the set of ground truths $\{f : TV_1[f] < \infty\}$ rather than just a minimax rate over $\{f : TV_1[f] \leq \alpha\}$ for some $\alpha \in \mathbb{R}$.

5. Empirically, AKORN is competitive with state of the art offline methods, even when they are artificially provided with the best possible hyperparameters.

### 1.1. Key Techniques and Other Contributions

1. We reduce knot-selection to optimal online denoising. By suitably measuring the stationarity of an online learner's predictions, AKORN adaptively generates a set of knots, which, when used to fit a linear regression spline, gives optimal statistical rates.

2. To this end, we introduce ADDLE (Adaptive Denoising with Linear Experts), which optimally solves the *online* non-parametric denoising problem for functions of bounded 1st-order TV: an extension of "Aligator" from (Baby et al., 2021).

3. We carry out the analysis for AKORN by introducing a fictitious estimator which allows us to deal with the statistical dependence between the adaptive knots and the data. This technique may be useful in other work on knot-selection.

## 2. Related Work

Regression over total variation classes is well-studied in the literature (Muller, 1992; Donoho & Johnstone, 1994; 1998; Tibshirani, 2014). Optimal techniques include wavelet smoothing (Donoho & Johnstone, 1994; 1998), Locally Adapive Regression Splines (Mammen & van de Geer, 1997), and the now state-of-the-art Trend Filtering estimate (Kim et al., 2009; Tibshirani, 2014). Crucially, most of these methods require an injection of a priori knowledge of the smoothness of $f$ via hyperparameter, unlike AKORN.

Some work makes use of Stein's Unbiased Risk Estimator (SURE) to select hyperparameters (Tibshirani & Taylor, 2012; Tibshirani, 2015; Donoho & Johnstone, 1995). These methods are typically heuristic, as extracting provable guarantees involves proving uniform convergence of SURE. The exception is (Donoho & Johnstone, 1995), which obtains this uniform convergence for wavelet smoothing. Though wavelets enjoy powerful adaptivity properties from a theoretical perspective, Trend Filtering achieves much better results in practice (Tibshirani, 2014).

There has also been interest in the online nonparametric regression setting, which surprisingly is not much harder than the offline setting (Rakhlin & Sridharan, 2014). Recently, Baby et al. (2021) uses oracle inequalities from the online literature (Hazan et al., 2006) to design an optimal parameter-free algorithm called "Aligator" for the online regression problem over $TV_0$. Unfortunately, when specialized to the offline setting, Aligator cannot compete with Trend Filtering empirically.

All minimax optimal estimates for TV classes must be nonlinear functions of the responses in order to display *local adaptivity*, as shown in Donoho & Johnstone (1998). For our method, this means knots must be adaptively spaced according to the change-points of the ground truth. Many algorithms have been proposed for knot selection (Friedman, 1991; Luo & Wahba, 1997; Wand, 2000), typically by choosing a large knotset and recursively purging knots (Goepp et al., 2025). To our knowledge, AKORN is the first knot selection algorithm with provable guarantees.

A more detailed discussion of related work is available in Appendix A.

## 3. Problem Setup

We now instantiate the model in Equation 1 by defining our assumption on $f$. We impose regularity on the ground truth $\boldsymbol{\theta} = [f(x_1), ...f(x_n)]$ as measured by its *1st-order total variation*.

**Definition 3.1.** The 1*st-order total variation* of a vector $\boldsymbol{\theta} \in \mathbb{R}^n$ with respect to the points $\mathcal{D}_X = \{x_1, ...x_n\}$ is defined as

$$TV_1[\boldsymbol{\theta}; \mathcal{D}_X] = \sum_{i=3}^{n} \left| \frac{\theta_i - \theta_{i-1}}{x_i - x_{i-1}} - \frac{\theta_{i-1} - \theta_{i-2}}{x_{i-1} - x_{i-2}} \right|$$

By extension, we define the discrete total variation of a function $f$ with respect to the points $\mathcal{D}_X = \{x_1, ..., x_n\}$ as

$$TV_1[f; \mathcal{D}_X] = TV_1[\boldsymbol{\theta}; \mathcal{D}_X]$$

where $\boldsymbol{\theta} = [f(x_1), ...f(x_n)]$.

As we shall be computing $TV_1$ with respect to fixed covariates, we tend to suppress $\mathcal{D}_X$ in the above notation.

*Remark* 3.2. The definitions adopted above, while standard (Guntuboyina et al., 2020), differ subtly from the definition of the true 1st-order total variation seminorm, $\| \cdot \|_{TV_1}$, defined for weakly differentiable functions (Tibshirani, 2014). To our knowledge, neither is more general than the other. However for $g$ differentiable, we have

$$TV_1[g; \mathcal{D}_X] \leq \|g\|_{TV_1}$$

for all $\mathcal{D}_X$, implying that $TV_1[f]$ can be replaced by $\|f\|_{TV_1}$ in our bound when $f$ is differentiable. Throughout this pa-

per, we use only the discrete total variation from Definition 3.1.

We assume that we are given data of the form $\{(x_i, y_i)\}_{i=1}^n$ according to Equation 1 where $f$ has bounded $TV_1$. We reserve the letter $C$ to denote $C := TV_1[f; \{x_i\}_{i=1}^n]$.

*Remark* 3.3. Why $TV_1[\cdot]$? While $TV_0$-functions can be approximated by a sparse combination of Heaviside functions, $TV_1$-functions are well approximated by linear splines with a sparse number of knots. Thus, in the $TV_0$ setting, it is "proper" to output a discontinuous, piecewise constant estimate ((Baby et al., 2023) provides a recipe for doing this with a sparse number of segments). On the other hand, "proper" estimates for $TV_1$ functions (and $TV_{k\geq 1}$) should be *continuous*, in addition to having a sparse number of change points. Thus, $TV_1$ is the first level at which the mismatch between inherently discontinuous online predictions and an inherently continuous ground truth must be addressed in offline-to-online reductions.

In the online setting, each data point comes to us one at a time, and the goal is to produce a sequence of predictions $\hat{y}_t$ so that $\sum_{t=1}^n (\hat{y}_t - \theta_t)^2$ is as small as possible. In Section 4.1, we describe an optimal parameter-free algorithm for the online problem.

In the offline setting (our main target), we wish to produce a model $\hat{f}$ such that $\sum_{i=1}^n (\hat{f}(x_i) - f(x_i))^2$ is small, assuming simultaneous access to all data points. Though the asymptotic rates for the offline setting and online setting are the same (up to lower-order terms) (Rakhlin & Sridharan, 2014; Baby & Wang, 2019), algorithms for the offline setting typically substantially outperform algorithms for the online setting. In Section 4.2, we propose a reduction from the offline setting to the online setting that mitigates the empirical drawbacks typically suffered by online algorithms.

### 3.1. Additional assumptions

We assume that $f$ is bounded by an unknown constant, $|f| \leq B$. As mentioned in (Baby et al., 2021), this assumption is typically not made in the literature (Donoho & Johnstone, 1994; Tibshirani, 2014). When $f$ is continuous, this assumption is vacuous. Without loss of generality, we let $B = 1$.

In the body of this paper, we assume that the covariates are equally spaced: $x_i = i/n$. This assumption is rather strong, but has been the starting point for many non-parametric regression algorithms, including Trend Filtering (Donoho & Johnstone, 1994; Tibshirani, 2014), where it was subsequently relaxed (Wang et al., 2014; Sadhanala & Tibshirani, 2019). In Appendix F, we show how a modified version of AKORN can handle uneven and random covariates.

## 3.2. Additional Notation

For a single natural number, $a$, we let $[a] := \{1, 2, ..., a\}$. For a real number, $z$, we let $(z)_+ = \max\{z, 0\}$. Let $e_i \in \mathbb{R}^n$ be the $i$th standard basis vector.

We introduce the notation $\mathcal{D}_X = \{x_1, ...x_n\}$ and $\mathcal{D}_Y = \{y_1, ...y_n\}$. When $k, k' \in \mathcal{D}_X$ with $k < k'$, we define $[k, k'] = \{x \in \mathcal{D}_X : k \leq x \leq k'\}$.

We also notate $\boldsymbol{\theta} = [\theta_1, ...\theta_n]^T = [f(x_1), ...f(x_n)]^T$. We assume that, for all $i \in [n]$, $x_i = i/n$. Strictly speaking, each $x_i$ is a scalar. However, as a matter of convenience, we will sometimes notate $[1, x_i]$ as $x_i$ when the distinction is clear from context. We define the truncated power basis, $\{g_i : [0, 1] \to \mathbb{R}\}_{i=1}^n$ as follows:

$$g_j(x) = (x - x_j)_+ \ ; \ 1 \leq i < n - 1$$

$$g_n(x) = 1$$

For any $i$, we vectorize the evaluations of $g_j$ on the data as $\boldsymbol{g}_j = [g_j(x_1), ...g_j(x_n)]^T$.

Finally, for any set of (non-repeating) knots $K = \{k_1, ...k_l\} \subset \mathcal{D}_X$, we let $\mathcal{G}(K) = \{\boldsymbol{g}_1, \boldsymbol{g}_n\} \cup \{\boldsymbol{g}_j\}_{j \in K}$. For this $K$, we also let $H_K$ be the matrix whose columns are $\{[1, (x_i - x_1)_+, g_{k_1}(x_i), ...g_{k_l}(x_i)]^T\}_{i=1}^n$. We use $S(K) = $ span $\mathcal{G}(K)$ to denote the space of (evaluations of) linear splines with knotset $K$. We use $F(K) = \text{span}\{\mathcal{G}(K) \cup \{\xi_1, ...\xi_{|K|}\}\}$, where $\xi_j = \sum_{i=j}^n e_i$ to denote the space of (evaluations of) piecewise linear functions with knotset $K$.

We sometimes abuse notation by identifying functions $p : [0, 1] \to \mathbb{R}$ with the finite-dimensional vector of their evaluations on $\mathcal{D}_X$, $[p(x_1), ..., p(x_n)]^T$. This is done only when the underlying function $p$ is clear from context, as is the case when we are discussing the ground truth ($f :=: \boldsymbol{\theta}$), or any linear spline ($\sum_{j=1}^n \beta_j g_j(x) :=: \sum_{j=1}^n \beta_j \boldsymbol{g_j}$).

# 4. ADDLE and AKORN

## 4.1. ADDLE: Online Denoising for $TV_1$

We first introduce ADDLE (ADaptive Denoising with Linear Experts) to treat the online problem of denoising the sequence of responses $\{y_1, ...y_n\}$ with a sequence $\{\hat{y}_1, ...\hat{y}_n\}$, where the responses come from the data model in Equation 1.

ADDLE operates by running Follow-the-Leading-History (FLH) (Hazan et al., 2006) with experts given by online linear regression[2] (Algorithm 3 in Appendix B) and loss functions $f_t(\cdot) = (\cdot - y_t)^2$. FLH predicts a weighted combination of the predictions by each expert, and uses an ex-

---

[2]Strictly speaking, ADDLE actually runs a clipped version of linear regression. See Appendix B for details

ponential reweighting scheme to update its weights at each time-step according to observed losses. A formal description of FLH/ADDLE appears in Appendix B.

Since our end goal is to address offline data, we assume the data is revealed in isotonoic order (i.e., our $t$-th observation is $y_t$), though this assumption can be relaxed by means of a geometric cover (Baby et al., 2021). Furthermore, the algorithm can be generalized to handle any $TV_k$ by augmenting the experts to perform online regression with $k$th degree polynomials.

*Remark* 4.1. Though ADDLE has not, to our knowledge, appeared in the literature, much of the technical scaffolding for online denoising over $TV$ classes via expert aggregation appeared in (Baby et al., 2021). As such, the *main* technical contribution of this paper is AKORN.

## 4.2. AKORN

As we have mentioned, one issue with ADDLE is that the predictions, $\{\hat{y}_i\}$, are not very useful in the offline setting. We now present AKORN, which uses online forward and backward passes together with an adaptive restarting rule in order to curate a set of knots, $K = \{k_1, ...k_l\} \subset \mathcal{D}_X$. With these knots in hand, AKORN then returns the best linear regression spline with knots in $K$:

$$\hat{f}(x) = [1, (x - x_1)_+, g_{k_1}(x), ...g_{k_l}](H_K H_K^T)^{-1} H_K Y$$

$$=: P_{S(K)} Y(x)$$

where we recall from Section 3.2 that $g_{k_i}$ is the truncated power basis function $(x - x_{k_i})_+$ and $H_K$ is the data-matrix whose columns are the features $[1, (\cdot - x_1)_+, (\cdot - k_1)_+, ...(\cdot - k_l)_+]^T$ and we have engaged in the aforementioned abuse of notation $\hat{f} = P_{S(K)} Y$.

To form $K$, AKORN begins by generating a forward knotset, $K_f$, and a backward knotset, $K_b$. These are generated by feeding the data to Algorithm 1 in isotonic and reverse-isotonic order respectively. Algorithm 1 is inspired by Algorithm 5 in (Baby et al., 2023), which was designed to impose a low-switching constraint on online predictions. Essentially, every time ADDLE starts (say, at time $b \in [n]$), the predictions of ADDLE are compared to a linear regression that also starts at time $b$. When the total square distance between these sequences drifts above a certain constant, we conclude that we have exited the interval in which we can linearly approximate $f$, and we put down a knot. The effect is that ADDLE restarts only when a noisy surrogate of the $TV_1$ within the interval exceeds $n/(\text{Interval Size})^{3/2}$ (see Lemma C.1).

To complete the construction of $K$, we generate preliminary fits $g = P_{F(K_f)} Y \in \mathbb{R}^n$ and $h = P_{F(K_b)} Y \in \mathbb{R}^n$ and form $\tilde{K}$, the set of all crossover points of $g$ and $h$. Crossover

**Algorithm 1** `FindKnots`

> **Input:** data $\{(x_t, y_t)\}_{t=1}^n$, variance $\sigma^2$
> $b \leftarrow 0$
> $\mathcal{K} \leftarrow \{\}$
> Start ADDLE instance $\mathcal{A}$
> **for** $t \in \{1, ..., n\}$ **do**
>  $\tilde{\theta}_t \leftarrow$ prediction for $x_t$ from $\mathcal{A}$
>  $\hat{a}_t \leftarrow$ `LinearLeastSquares`$(z_{b:(t-1)})$ $\{\in \mathbb{R}^2\}$
>  For all $j \in [b, ...t]$ set $\hat{w}_j^t \leftarrow \hat{a}_t^T x_j$
>  $s \leftarrow \sum_{j=b}^t (\hat{w}_j^t - \tilde{\theta}_t)^2$
>  **if** $s > 5\sigma^2 \log \frac{2n^2}{\delta}$ **then**
>    $\mathcal{K} = \mathcal{K} \cup \{x_{t-1}\}$
>    $b = t$
>    Restart $\mathcal{A}$
>  **end if**
>  Update $\mathcal{A}$ with $y_t$
> **end for**
> output $\mathcal{K}$

---

**Algorithm 2** AKORN

> **Input:** data $\{x_t, y_t\}_{t=1}^n$, variance $\sigma^2$
> $K_f \leftarrow$ `FindKnots`$(\{x_t, y_t\}_{t=1}^n, \sigma^2)$
> $K_b \leftarrow$ `FindKnots`$(\{x_t, y_t\}_{t=n}^1, \sigma^2)$
> $g \leftarrow P_{F(K_f)} Y$
> $h \leftarrow P_{F(K_b)} Y$
> $\tilde{K} \leftarrow$ `Crossovers`$(g, h, \{x_t\}_{t=1}^n)$
> $K_{collision} \leftarrow \{x_t : (t < n-1) \wedge x_{t+1} \in K_f \cap K_b\}$
> $K = K_f \cup K_b \cup \tilde{K} \cup K_{collision}$
> output $\hat{f} = P_{S(K_f \cup K_b \cup \tilde{K})} Y$

---

points are defined as covariates $z_t$ where either of the following holds

1. $h(z_t) \geq g(z_t)$ and $g(z_{t+1}) > h(z_{t+1})$

2. $g(z_t) \geq h(z_t)$ and $h(z_{t+1}) > g(z_{t+1})$

We then report $K = K_f \cup K_b \cup \tilde{K}$ and perform least-squares regression of $Y$ onto the space $S(K)$.

In effect, AKORN is forced to first think about the data from an online perspective – at this step it forms a qualitative understanding of the ground truth $f$, summarized by the knot sets $K_f$ and $K_b$. It then combines this understanding with offline access to the data in order to produce a fit that is simultaneously attractive and adaptive.

*Remark* 4.2. Strictly speaking, our proofs require that we also add to $K$ all points in the set $K_{collision} := \{x_t : (t < n-1) \wedge x_{t+1} \in K_f \cap K_b\}$. When $n$, the number of data points, is large and $\sigma > 0$, it is rare that $K_f$ and $K_b$ share knots, so we mention this step only parenthetically.

### 4.3. Computational Complexity

Using an $O(1)$-time update rule for each linear expert, AD-DLE can be implemented in $O(n^2)$ time. AKORN also runs in $O(n^2)$. As an aside, we observe that Algorithm 1 can be viewed as an optimized form of ADDLE that adaptively purges the pool of experts. Using a Geometric Cover as in (Baby et al., 2021), it is straightforward to reduce the run-time of ADDLE to $O(n \log n)$. However, this does not immediately lead to an $O(n \log n)$ runtime for AKORN.

For comparison, the worst-case computational complexity of Trend Filtering is $O(n^{3/2} \log \frac{1}{\epsilon})$ to find an $\epsilon$-approximate solution (Tibshirani, 2014). This does not take into account the cost of parameter tuning. If we tune parameters heuristically using Stein's Unbiased Risk Estimator at a discretization level $\Delta$, we need to solve trend filtering $C/\Delta$ times and compute the the effective degrees of freedom (dof) of each fit. In general, the only possible a-priori upper bound on $C$ is $C = O(n^2)$, leading to a generic complexity of $O(n^{7/2} \log 1/\epsilon)$. In practice, algorithms for solving trend filtering are extremely fast, and the main computational burden comes from computing dof for several candidate fits.

## 5. Experimental Results

### 5.1. Local adaptivity

As a matter of interest, we begin by noting that the fitted values from ADDLE can be expressed as

$$\hat{Y}_{addle} = W_{addle} Y$$

for some hat-matrix $W_{addle}$ that depends on the weights of the ADDLE instance. We can do the same for AKORN:

$$\hat{Y}_{akorn} = W_{akorn} Y$$

where $W_{akorn} = H_K^T (H_K H_K^T)^{-1} H_K$. Because the bandwidth of these matrices around each diagonal element $(i, i)$ corresponds to the neighborhood of the data used in forming the $i$th prediction, $\hat{y}_i$, we dub $W_{addle}$ and $W_{akorn}$ "attention maps."

It is informative to compare these learned attention maps to the static attention induced by local linear regression, as we do in Figure 1 for Donoho & Johnstone (1994)'s "Doppler" function. Unlike local linear regression, we can see that that ADDLE has learned how to adaptively optimize a bias-variance trade-off in the neighborhood of each data point by choosing a spatially varying "bandwidth". Similarly, we see that AKORN inherits ADDLE's learned knowledge of the geometry of the ground truth. It is well-known that this kind of local adaptivity is essential to getting optimal rates over $TV$ classes (Donoho & Johnstone, 1994).

**Empirical Rates**

|  | PW-Lin | Doppler | Jump |
|---|---|---|---|
| Oracle TF | -0.95 | -0.84 | -0.37 |
| AKORN | -1 | -0.91 | -0.7 |

*Table 1.* Estimated rates as determined as the slopes of the lines corresponding to Oracle TF and AKORN in Figure 2. Each entry in the table gives the exponent, $\gamma$, in the rate $O(n^\gamma)$

**MSEs for Doppler Function (n = 1000)**

| $\sigma$ | Oracle TF | DoF TF | AKORN | Wavelets |
|---|---|---|---|---|
| 0 | 0 | 0 | 0 | 0 |
| 0.1 | 0.0004 | 0.0006 | 0.0008 | 0.005 |
| 0.2 | 0.0012 | 0.0016 | 0.0023 | 0.014 |
| 0.3 | 0.0023 | 0.003 | 0.0039 | 0.024 |
| 0.4 | 0.0034 | 0.0041 | 0.0052 | 0.034 |
| 0.5 | 0.0057 | 0.007 | 0.007 | 0.046 |

*Table 2.* Values averaged over 20 runs and rounded to the nearest $10^{-4}$

## 5.2. Performance comparison

While ADDLE's ability to learn the local smoothness of the ground truth is remarkable, it performs comparatively poorly on offline datasets, as we demonstrate in Figure 5 in Appendix G.2. In this section, we show that AKORN is able to use ADDLE's adaptivity while efficiently leveraging offline data. We compare several policies.

1. *Oracle linear trend filtering*. We solve the variational problem described in Section A for a grid of possible $\lambda$. We then measure the true MSE against the ground truth, and return the best fit. We emphasize that this policy requires oracle knowledge of the ground truth.

2. *DoF linear trend filtering*. We solve the same trend filtering optimization problem for the same set of possible $\lambda$. We then form the Stein estimate of the risk by estimating the degrees of freedom of each model (Tibshirani & Taylor, 2012), and choose the best fit. Note that this procedure has no theoretical guarantees, is computationally intensive, and can require high-precision arithmetic when the dataset is large.

3. *Wavelets*. We use the soft-thresholding estimator of (Donoho & Johnstone, 1998) with Debauchies 2 wavelets. To our knowledge, apart from AKORN, this is the only optimal and parameter-free algorithm for estimating $TV_1$ functions.

4. *AKORN*. We run AKORN as described in Section 4.2, with failure probability $\delta = 0.1$.

In Figure 2, we display a log-log plot of the error of each policy for various ground truths, against exponentially increasing values of $n$. In Table 1, we report the slope of the lines corresponding to Oracle Trend Filtering and AKORN, as estimated by the Linear Least Squares fit. This gives the approximate rate of each estimator. Across all ground truths, we observe that AKORN competes closely with Trend Filtering, despite the fact that the latter is provided with access to the ground truth.

The first example in Figure 2, together with the first column in Table 1, suggests that AKORN adaptively achieves the parametric rate $\frac{1}{n}$ on piecewise linear functions, as does trend filtering. The second example in Figure 2 and

Table 1 validates that AKORN achieves the claimed rate of $\tilde{O}(n^{-4/5})$ on spatially heterogenous functions like the Doppler function of (Donoho & Johnstone, 1994).

The final example in Figure 2 represents runs of each policy on the ground truths $\boldsymbol{\theta}_n = [\mathbf{0}^T \in \mathbb{R}^{1 \times n-5}, 1, 2, 3, 4, 5]^T$. In this setting, the fast rates for Trend Filtering from (Guntuboyina et al., 2020) do not apply, because the final linear segment of the ground truth is small. Notice that in this case, $TV_1[\boldsymbol{\theta}] = \Theta(n)$, which means that the rate predicted both by our theory and that of Trend Filtering is $\tilde{O}(n^{-4/5}n^{2/5}) = \tilde{O}(n^{-2/5})$. While this rate seems to be accurate for Oracle Trend Filtering, our experimental results seem to indicate that AKORN outperforms this rate substantially, as the least-squares slope of the orange line is about $-0.7$ (Table 1). These results indicate that AKORN's enhanced adaptivity leads to favorable performance on highly irregular problem instances. Visually, we see that AKORN's proposed model is much more attractive than that of Trend Filtering.

In Table 2, we report the MSEs of each policy for fixed $n$ and various noise levels $\sigma$. From the table, we gather that AKORN is competitive with both Oracle and DoF Trend Filtering, especially for larger values of $\sigma$. Wavelets is substantially behind all other policies.

In summary, AKORN's performance is very close to both Oracle Trend Filtering and DoF Trend Filtering across all tests, and it outperforms even Oracle Trend Filtering on such pathologies as the jump function of Figure 2. Wavelet denoising, which is the only other method we know of that provides adaptivity to $TV_1[f]$, is behind the pack empirically. Code is available at github.com/SunilMadhow/AKORN.

## 6. Theoretical Results

We begin by confirming that ADDLE achieves the optimal total square error, $\tilde{O}(n^{1/5}C^{2/5})$. Note that this implies that the average square error (nearly) matches the optimal rate, $\tilde{O}(n^{-4/5}C^{2/5})$

**Theorem 6.1** (Bound on ADDLE error). *Consider equally*

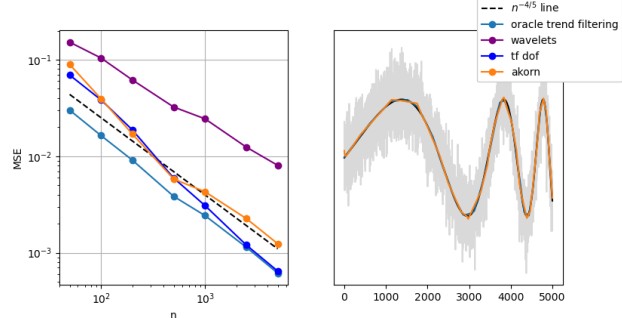

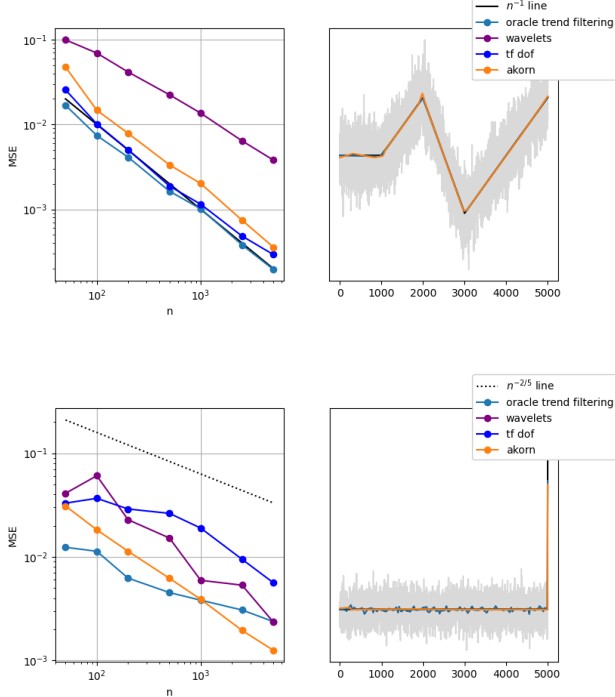

*Figure 2.* Estimated rates for AKORN and Oracle Trend Filtering formed using 20 Monte-Carlo runs for each $n$, together with representative fits for $n = 5000$. $\sigma = 0.3$, $\delta = 0.1$. Top left: PW Linear function; Top Right: Doppler function; Bottom: Jump function.

*spaced design points $\{x_i\}_{i=1}^n$ and any $f$ with $C := TV_1[f; \mathcal{D}_X] < \infty$. Let responses $\{y_t\}$ come from the model in Equation 1. Let $\{\hat{y}_t\}_{t=1}^n$ be the the predictions generated by ADDLE when fed the data in isotonic order. With probability $1 - \delta$, the total squared error satisfies:*

$$\sum_{t=1}^n (\hat{y}_t - f(x_t))^2 = \tilde{O}(n^{1/5} C^{2/5})$$

*where $\tilde{O}$ hides constants (including $\sigma$) and polylog factors of $n$ and $\delta$.*

With this result in hand, we are able to prove that the function, $\hat{f}$, outputted by AKORN also adaptively achieves the rate of $\tilde{O}(n^{-4/5} C^{2/5})$.

**Theorem 6.2** (Bound on AKORN MSE). *Consider equally spaced design points, $\{x_t = t/n\}_{t=1}^n$ and $f$ such that $C := TV_1[f; \mathcal{D}_X] < \infty$. Let responses $\{y_t\}$ come from the model in Equation 1. Let $\hat{f}$ be the function returned by AKORN. Then, with probability $1 - \delta$, the average square error satisfies:*

$$\frac{1}{n}\sum_{t=1}^n (\hat{f}(x_t) - f(x_t))^2 = \tilde{O}(n^{-4/5} C^{2/5})$$

*where $\tilde{O}$ hides constants (including $\sigma$) and polylog factors of $n$ and $\delta$.*

Let us emphasize the message of Theorem 6.2 by comparing it with the standard guarantees of Locally Adaptive Regression Splines and Trend Filtering (Donoho & Johnstone, 1998; Mammen & van de Geer, 1997; Tibshirani, 2014), each of both of which provide an algorithm $\mathcal{A}_\lambda$ with hyperparameter $\lambda$ so that for any $\alpha \in \mathbb{R}$, there exists $\lambda(\alpha)$ so that $A_{\lambda(\alpha)}$ (nearly) achieves the minimax rate $O(\alpha^{2/5} n^{-4/5})$ over $\{f : \|f\|_{TV_1} \leq \alpha\}$.

On the other hand, Theorem 6.2 says that AKORN is not only (nearly) minimax over $\{f : TV_1[f] \leq \alpha\}$ in a parameter-free way, but it achieves the *instance-dependent* rate $\tilde{O}(TV_1[f]^{2/5} n^{-4/5})$ over $\{f : TV_1[f] < \infty\}$.

In Appendix F, we describe how a modified version of ADDLE can achieve the same rate (up to log terms) when the covariates $\{x_i\}_{i=1}^n$ satisfy $\max_{i=2,\ldots,n} |x_i - x_{i-1}| \leq \frac{\log n}{p_0 n}$. This condition is satisfied with high probability when $x_i \overset{iid}{\sim} p_X(\cdot)$, where $p_X$ is a density with support on $[0, 1]$ that satisfies $p_X(x) \geq p_0 > 0$ for $x \in [0, 1]$ (Wang et al., 2014). This implies an optimal variant of AKORN under the same conditions. Detailed theorem statements and proofs can be found in Appendix F.

## 7. Proof Sketches

### 7.1. Proof Sketch of Theorem 6.1

The proof of Theorem 6.1 follows along the lines of (Baby et al., 2021). The crucial component is the following classical Lemma from (Hazan & Seshadhri, 2007).

**Proposition 7.1** ((Hazan & Seshadhri, 2007) informal). *For any interval $I = [r, s]$ in time, the algorithm FLH (Fig.4) with learning rate $\zeta = \alpha$ applied to the loss functions $\ell_t(\cdot) = (\cdot - y_t)^2$ gives $O(\alpha^{-1}(\log r + \log |I|))$ regret against the best base learner in hindsight, where $\alpha$ upper bounds the parameter of exp-concavity for all $\ell_t$.*

For any fixed partition $\mathcal{P}$ of $[n]$ into intervals, this Lemma

directly controls the quantity

$$\sum_{p=[r,s]\in\mathcal{P}}\sum_{t\in p}(\hat{y}_t-y_t)^2-(\hat{z}^r(t)-y_t)^2 \qquad (2)$$

where $\hat{z}^r(t)$ is the prediction at time $t$ of an online linear regression expert that starts at time $r$ (Specifically, an instance of Algorithm 3 in Appendix B).

Appendix D concerns itself with establishing:

1. The existence of a partition, $\mathcal{P}^*$, of $f$ into $O(n^{1/5}C^{2/5})$ roughly linear chunks, such that Equation (2) is $\tilde{O}(1)$ for each $p\in\mathcal{P}^*$.

2. A statistical control on the difference between Equation 2 and the corresponding quantity with noisy responses $y_t$ replaced by the ground truth $\theta_t=f(x_t)$.

One appealing aspect of such proofs is that the heavy lifting is done by approximation theoretic analysis. That is, we can encode oracle knowledge about the structure of $f$ into the partition $\mathcal{P}^*$ while proving theorems, and be sure that FLH will discover this structure without any additional algorithmic input. This type of adaptivity is the core contribution that online learning can make to statistics. The complete proof for ADDLE is in Appendix D

### 7.2. Proof Sketch of Theorem 6.2

In order to prove the optimality of $\hat{f}=P_{S(K)}Y$, we begin by establishing properties about the knotset $K$. Recall that $K=K_f\cup K_b\cup\tilde{K}$, where $K_f$ and $K_b$ are generated according to the online passes of Algorithm 1. Using the optimality of ADDLE (Theorem 6.1), we can prove the following "Change-point Detection Lemma," which tells us that $K_f$ (and $K_b$) divide $f$ into a small number of piecewise linear chunks.

**Lemma 7.2** (Change-point Detection Lemma (Lemma C.1 informal)). *With high probability, we have*

$$|K_f|\lesssim\max\{n^{1/5}C^{2/5},1\}$$

*and for all $k_i^f\in K_f$ there is a linear function $w^i$ defined on $[k_i^f,k_{i+1}^f)$ such that*

$$\sum_{t=k_i^f}^{k_{i+1}^f-1}(w^i(x_t)-\theta_t)^2\lesssim 1+n_i^{1/5}TV_1[\theta^i]^{2/5}$$

While the adaptivity an online algorithm is typically evaluated on the basis of its regret bound, the Change-point Detection Lemma tells us that online algorithms are implicitly making fairly deep inferences about their data. In

particular, ADDLE provides not only predictions with small error, but access to a sparse set of change-points that encode an optimal bias/variance tradeoff around each covariate.

The properties established in the Change-point Detection Lemma quickly imply that back-fitting piecewise linear functions on either $K_f$ or $K_b$ gives the optimal rate. In particular, the smallness of $K_f$ (resp. $K_b$) allows us to control the variance of $P_{F(K_f)}Y$ (resp. $P_{F(K_b)}Y$), while the approximate linearity property allows us to control $\|\mathbb{E}[P_{F(K_f)}Y]-\boldsymbol{\theta}\|_2^2$ (resp. $\|\mathbb{E}[P_{F(K_b)}Y]-\boldsymbol{\theta}\|_2^2$) (Lemmas C.3, C.4 and C.5 in the Appendix). This certifies that $g=P_{F(K_f)}Y$ and $h=P_{F(K_b)}$ have MSE scaling as $\tilde{O}(n^{1/5}C^{2/5})$. While $g$ and $h$ are, in general, discontinuous (and therefore improper estimates of the ground truth $f$), we rely on their existence later in the proof. We summarize in the following Lemma:

**Lemma 7.3.** *[Corollary C.6 (informal)] With high probability, we have*

$$\|P_{F(K_f)}Y-\boldsymbol{\theta}\|_2^2=\tilde{O}(n^{1/5}C^{2/5})$$

*and*

$$\|P_{F(K_b)}Y-\boldsymbol{\theta}\|_2^2=\tilde{O}(n^{1/5}C^{2/5})$$

Turning our attention to $\hat{f}=P_{S(K)}Y$, Lemma C.2 provides us with the following bound, which relates the square error of $\hat{f}=P_{S(K)}Y$ with that of a fictitious estimator $\hat{f}_f=P_{F(K)}Y$.

$$\|P_{S(K)}Y-\boldsymbol{\theta}\|_2^2\leq 2\|P_{F(K)}Y-\boldsymbol{\theta}\|_2^2+2\|P_{S(K)}\boldsymbol{\theta}-\boldsymbol{\theta}\|_2^2 \tag{3}$$

This is crucial because we are able to cover the space of possible knot sets $K$ in the first term on the right hand side by covering intervals independently. The first term is easily bounded using the methodology of Lemma 7.3. The second term is free from dependence on the responses, $Y$, and is bounded using the following approximation theoretic lemma, which asserts the existence of a linear spline $s\in S(K_f\cup K_b\cup\tilde{K})$ whose curve lies in between the curves of any two functions $g\in F(K_f)$ and $h\in F(K_b)$.

**Lemma 7.4.** *[Lemma C.7 (Informal)] If $K_f\cap K_b=\{\}$ then for all $g\in F(K_f)$ and $h\in F(K_b)$ there exists $s\in S(K)$ so that for all $x\in\mathcal{D}_X$ there exists $\lambda_x\in[0,1]$ with*

$$s(x)=\lambda_x g(x)+(1-\lambda_x)h(x)$$

Finally, we apply Lemma 7.4 with $g=P_{F(K_f)}Y$ and $h=P_{F(K_b)}Y$, in order to assert the existence of $s\in S(K)$ with small error. Concretely, by the convexity of square loss, we have in Equation 3

$$\|P_{S(K)}\boldsymbol{\theta}-\boldsymbol{\theta}\|_2^2\leq\|s-\boldsymbol{\theta}\|_2^2\leq$$

$$\|g-\boldsymbol{\theta}\|_2^2+\|h-\boldsymbol{\theta}\|_2^2=\tilde{O}(n^{1/5}C^{2/5})$$

in Equation 3. The complete proof for Theorem 6.2 is in Appendix C.

## 8. Conclusion and Future Work

The main contribution of this paper is AKORN, a parameter-free algorithm which uses ideas from online learning to produce a function $\hat{f}$ that (a) empirically competes with the output of linear Trend Filtering, even when the latter is given oracle access to the ground truth for hyperparameter tuning (b) achieves the optimal rate for $TV_1$-bounded functions by adapting to the local smoothness of $f$ (c) operates by means of a reduction from knot-selection to adaptive online prediction.

A major limitation of AKORN is that it does not handle higher order $TV_k$ classes. An extension to arbitrary $k > 1$ would place AKORN on even footing with Trend Filtering theoretically. They key challenge here is generalizing Lemma 7.4 to splines of higher degree. Another limitation of AKORN is its $O(n^2)$ runtime. We believe that a more careful reduction to a geometric cover version of ADDLE (Baby et al., 2021) may lead to an $O(n \log n)$ algorithm, though in practice this may come at the cost of additional MSE.

Online methods have long been promising to expand the scope of theory by eliminating assumptions on optimally tuned parameters and enhancing adaptivity to problem instances (Cutkosky & Orabona, 2018; Cutkosky et al., 2023). If, as AKORN suggests, such methods can be modified to compete ex-situ with state-of-the-art *offline* algorithms, offline-to-online reductions could yield new theorems whose hypotheses are more likely to hold in real-world scenarios.

## Acknowledgments

We thank the anonymous reviewers for their thoughtful feedback, which helped improve the clarity and presentation of this paper. This work was partially supported by NSF Awards #2134214 and #2007117.

## Impact Statement

This paper presents work whose goal is to advance the field of Machine Learning. There are many potential societal consequences of our work, none which we feel must be specifically highlighted here.

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

    (a) Player plays an action $z_t \in V$

    (b) Universe chooses a loss function $f_t$

    (c) Player suffers loss $f_t(z_t)$

*Figure 3. Online interaction protocol*

## A. More Background on Related Work

One solution to the non-parametric problem of Equation 1 is given by *k-th order trend filtering* (Kim et al., 2009; Tibshirani, 2014), which efficiently solves the minimization problem

$$\hat{f}_{tf} = \arg\min_{g \in \mathcal{U}_n^k} \sum_{i=1}^n (y_i - g(x_i))^2 + \lambda TV_k(g) \tag{4}$$

where $\mathcal{U}_n^k$ is the span of a certain collection of functions called the *falling factorial* basis functions (Wang et al., 2014; Tibshirani, 2022). For $\lambda = \Phi(n^{\frac{1}{2k+3}} C^{\frac{-(2k+1)}{2k+3}})$, the loss of $\hat{f}_{tf}$ satisfies

$$\frac{1}{n}\sum_{i=1}^n (f(x_i) - \hat{f}(x_i))^2 = \tilde{O}_P(n^{\frac{-(2k+2)}{(2k+3)}} C^{\frac{2}{2k+3}})$$

where $C$ is an upper bound on $TV_k(f)$, which is known to be the minimax-optimal rate. Crucially, in order to benefit from tight bounds, a practitioner would need to choose $\lambda$ with knowledge of the smoothness of the the ground truth, $f$. Said another way, Trend Filtering is optimal only over the class of functions $\mathcal{F}_k(C) = \{f : TV_k(f) \leq C\}$ and requires knowledge of $C$. For a comprehensive treatment of the theory underlying Trend Filtering, which is the state of the art solution for regression over $TV_k$ classes, we refer the reader to (Tibshirani, 2022).

Regression over non-parametric classes is a much broader field, with a well-developed theory of minimaxity (Donoho & Johnstone, 1994; Mammen & van de Geer, 1997; Rakhlin & Sridharan, 2014; Tibshirani, 2014; Baby & Wang, 2019). In particular, (Rakhlin & Sridharan, 2014) observes that the complexity of the online problem typically does not differ too much from that of the offline problem.

$TV_k$ function classes admit functions whose smoothness varies spatially, which makes the estimation problem especially challenging. While Holder or Sobolev functions can be optimally estimated by linear smoothers, optimal estimators for $TV_k$ must display *local adaptivity* to the smoothness of the ground truth (Muller, 1992; Donoho & Johnstone, 1994; Mammen & van de Geer, 1997; Tibshirani, 2014).

### A.1. Online learning

Online learning studies algorithms for playing the game in Figure 3.

The goal is to choose actions such that the cumulative loss is small (with respect to some comparator). This setting is entirely non-stochastic, and we therefore measure an algorithm's performance in terms of the regret of its predictions against a comparator:

$$\text{Regret}(z_1, ...z_n | u) = \sum_{t=1}^T (f_t(z_t) - f_t(u))$$

If we let $f_t(\cdot) = (\cdot - y_t)^2$, we have a stochastic relaxation of the adversarial setting. When specialized to stochastic/batch settings, OL algorithms often enjoy remarkable adaptivity to problem features (Orabona, 2014; Baby et al., 2021; Wu et al., 2021; Cutkosky et al., 2023). In our setting, particularly relevant is the Aligator algorithm (Baby et al., 2021), which addresses online nonparametric denoising over $TV_0$ by using an expert aggregation algorithm to adaptively select the best window in which to perform averaging for each $x_t$. For $k = 0$, Aligator's predictions incur the optimal error of $\tilde{O}(n^{1/3} C^{1/3})$.

Central to our work is the theory of adaptivity in OL (Hazan & Seshadhri, 2009; Daniely et al., 2015), which seeks to guarantee that the regret within *each interval* is small with respect to some set of comparators. In particular, we will make use of the Follow-the-Leading-History expert aggregation algorithm from (Hazan et al., 2006), which has the remarkable property that, in any interval, the aggregate prediction competes with each individual expert.

Online forecasting over $TV_k$-bounded sequences has also been addressed for $k = 0, 1$ in the fully adversarial setting in a sequence of papers by (Baby & Wang, 2021; Baby et al., 2021; Baby & Wang, 2023).

More broadly, it is widely believed that hyperparameters, which can typically only be tuned heuristically, are responsible for large gaps between learning theory and practice across the entire discipline of Machine Learning (Chaudhuri et al., 2009). This has led to a growing movement to design *parameter-free* algorithms, whose theoretical guarantees can reasonably be expected to hold in real-world scenarios (Cutkosky & Orabona, 2018; Chaudhuri et al., 2009; Orabona, 2014).

## B. Algorithm descriptions

The following algorithm, due to Hazan & Seshadhri (2007), is called Follow-the-Leading-History (FLH). ADDLE operates by running FLH with online linear regression experts.

---

FLH: inputs - Learning rate $\zeta$ and $n$ base learners $E^1, \ldots, E^n$

1. For each $t$, $v_t = (v_t^{(1)}, \ldots, v_t^{(t)})$ is a probability vector in $\mathbb{R}^t$. Initialize $v_1^{(1)} = 1$.

2. In round $t$, set $\forall j \leq t$, $x_t^j \leftarrow E^j(t)$ (the prediction of the $j^{th}$ bas learner at time $t$). Play $x_t = \sum_{j=1}^{t} v_t^{(j)} x_t^{(j)}$.

3. After receiving $f_t$, set $\hat{v}_{t+1}^{(t+1)} = 0$ and perform update for $1 \leq i \leq t$:

$$\hat{v}_{t+1}^{(i)} = \frac{v_t^{(i)} e^{-\zeta f_t(x_t^{(i)})}}{\sum_{j=1}^{t} v_t^{(j)} e^{-\zeta f_t(x_t^{(j)})}} \tag{5}$$

4. Addition step - Set $v_{t+1}^{(t+1)}$ to $1/(t+1)$ and for $i \neq t+1$:

$$v_{t+1}^{(i)} = (1 - (t+1)^{-1}) \hat{v}_{t+1}^{(i)} \tag{6}$$

---

*Figure 4.* FLH algorithm (copied verbatim from (Baby & Wang, 2021))

In the course of the paper, we will make references to both bounded linear regression experts and unbounded linear regression experts, described in Algorithms 3 and 4 respectively.

Algorithm 5 gives the full description of ADDLE, which technically requires us to use the bounded linear regression expert.

---

**Algorithm 3** Bounded Linear Regression Expert

---

**Input**: history $\mathcal{H} \subset (\mathcal{X} \times [0,1])^*$, feature $x$, bound $B$ {Generates $B$-bounded prediction given history $\mathcal{H}$}
**if** $\mathcal{H}$ is empty **then**
    Output 0
**else if** $\mathcal{H} = \{(x_1^{\mathcal{H}}, y_1^{\mathcal{H}})\}$ **then**
    Output $[y_1^{\mathcal{H}}]_{[-B,B]}$ {$z_{[A,B]}$ clips a real number $z$ between $A$ and $B$}
**else**
    $\hat{l} \leftarrow$ linear least squares fit on $\mathcal{H}$
    Output $\hat{l}(x)_{[-B,B]}$
**end if**

---

---

**Algorithm 4** Linear regression expert

---

    **Input**: history $\mathcal{H} \subset (\mathcal{X} \times [0,1])^*$, feature $x$ {Generates prediction given history $\mathcal{H}$}
    **if** $\mathcal{H}$ is empty **then**
        Output 0
    **else if** $\mathcal{H} = \{(x_1^{\mathcal{H}}, y_1^{\mathcal{H}})\}$ **then**
        Output $y_1^{\mathcal{H}}$
    **else**
        $\hat{l} \leftarrow$ linear least squares fit on $\mathcal{H}$
        Output $\hat{l}(x)$
    **end if**

---

---

**Algorithm 5** ADDLE

---

    **input** $\mathcal{D}, \sigma$
    $B \leftarrow \max_i |y_i| + \max\{\sigma\sqrt{2\log 4n/\delta}, 1\}$
    Run FLH with learning rate $\eta = \frac{1}{8(1+\sigma\sqrt{\log 2n/\delta})^2}$, base learners $E^j(t) = \texttt{Predict}(\{(x_k, y_k)\}_{k=j}^{t-1}, x_t, B)$

---

## C. Proofs for AKORN

### C.1. Properties of Knot Selection

For $\boldsymbol{\theta}$ fixed, and some interval $I = [r, s] \subset [n]$, we introduce the notation $TV_1(I) := TV_1[\boldsymbol{\theta}[r:s])]$ In the following lemma, we prove that Algorithm 1 induces a not-too-large partition of $f$ into roughly linear segments.

**Lemma C.1** (Compare to Theorem 18 in (Baby et al., 2023)). *Consider the set of knots, $K_0 = \{k_1, ...k_l, k_{l+1} := x_{n+1}\}$, outputted by Algorithm 1 when exposed to data isotonically, as well as the induced partition*

$$\mathcal{P} = \{p_i := [k_i, ...k_{i+1} - 1]| \ i \in [l]\}$$

*Let $n_i$ be the size of $p_i$.*

*Then, for any $\delta > 0$, there is an event $\mathcal{E}_1(\delta)$ which holds with probability at least $1 - \delta$ and upon which*

$$|\mathcal{P}| \lesssim \max\{n^{1/5}C^{2/5}, 1\}$$

*and*

$$\forall i \ \exists \ w^i \ \text{linear} \ : \ \sum_{t \in p_i}(w^i(x_t) - \theta_t)^2 \lesssim 1 + n_i^{1/5}TV_1(p_i)^{2/5}$$

*where $\lesssim$ hides constants and polylog factors of $\delta$, n.*

*Proof.* We follow the methodology of Theorem 18 from (Baby et al., 2023). Suppose that for time $t$, line the condition inside the if-statement of Algorithm 1 is executing for the $i$th time. We start by bounding $C_i := TV_1([b, t])$.

$$\sum_{j=b+1}^{t}(\hat{w}_j^t - \tilde{\theta}_j)^2 \leq \sum_{j=b+1}^{t}(\hat{w}_j^t - \mathbb{E}[\hat{w}_j^t] + \mathbb{E}[\hat{w}_j^t] - \theta_j + \theta_j - \tilde{\theta}_j)^2$$

$$\leq 2\sum_{j=b+1}^{t}(\hat{w}_j^t - \mathbb{E}[\hat{w}_j^t])^2 + 4\sum_{j=b+1}^{t}(\mathbb{E}[\hat{w}_j^t] - \theta_j)^2 + 4\sum_{j=b+1}^{t}(\theta_j - \tilde{\theta}_j)^2 \tag{7}$$

By Lemma C.10, the first term is bounded by $4\sigma^2 \log(1/\delta)$ with probability $1-\delta$. The second term is bounded by $C_i^2 n_i^3/n^2$, which can be shown by direct computation, as in Appendix E. By Theorem D.1 (Optimality of ADDLE), the final term is bounded (with probability $1 - \delta$) by $\iota n_i^{3/5} C_i^{2/5}/n^{2/5}$, where $\iota$ contains constants and log-factors. Union bounding and then combining this with the condition inside the if statement, we see that with probability $1 - \delta$ we have:

$$4\sigma^2 \log\left(4/\delta\right) + 4\max\{\iota C_i^{2/5} n_i^{3/5}/n^{2/5}, C_i^2 n_i^3/n^2\} \geq 5\sigma^2 \log 4/\delta \tag{8}$$

Regardless of which value the maximum takes, we conclude that

$$C_i \gtrsim n/n_i^{3/2} \tag{9}$$

At the same time, on this same event, we have by definition that

$$\sum_{j=b+1}^{t} (\hat{w}_j^t - \theta_k)^2 \leq 2 \sum_{j=b+1}^{t} (\hat{w}_j^t - \tilde{\theta}_k)^2 + 2 \sum_{j=b+1}^{t} (\tilde{\theta}_j - \theta_j)^2 \lesssim 1 + n_i^{1/5} C_i^{2/5}$$

We may now cover all $n^2$ possible intervals, which amounts to adding an $n^2$ in the log and yields $\mathcal{E}_1(\delta)$.

Now let $M - 1$ be the total number of times that the if-block is executed. So $M = |\mathcal{P}|$ is the number of bins spawned. Returning to Equation 9, Jensen's inequality with $\phi : x \mapsto x^{-3/2}$ gives

$$C \geq \sum_{i=1}^{M-1} C_i$$

$$\gtrsim \sum_{i=1}^{M-1} \frac{n}{n_i^{3/2}}$$

$$= \sum_{i=1}^{M-1} \phi\left(\frac{n_i}{n^{\frac{2}{3}}(M-1)^{\frac{2}{3}}}\right) \frac{1}{(M-1)}$$

$$\geq \phi\left(\sum_{i=1}^{M-1} \frac{n_i}{n^{\frac{2}{3}}(M-1)^{\frac{5}{3}}}\right)$$

$$\geq \phi(n^{\frac{1}{3}}/(M-1)^{\frac{5}{3}}) = (M-1)^{\frac{5}{2}}/n^{\frac{1}{2}}$$

So for $M > 1$, we have

$$|\mathcal{P}| = M \lesssim M - 1 \lesssim C^{2/5} n^{1/5}$$

$\square$

## C.2. Introducing the fictitious estimator: $P_{S(K)}Y$ versus $P_{F(K)}Y$

In our analysis, it is necessary to introduce a fictitious estimator that conducts independent fits within each partition. This will allow us to cover knot-sets more efficiently. The following lemma relates the error of $\hat{f}$ to the fictitious estimator, $\hat{f}_f$. Recall the definitions of $F(K)$ and $S(K)$ for knotsets $K$ from Section 3.2. Note that, in the following lemma, we abuse notation by identifying $\hat{f}$ with $[\hat{f}(x_1), ...\hat{f}(x_n)]^T$ (and likewise for $\hat{f}_f$), which allows us to use the notation $P_{S(K)}f(z)$ for $[1, (z - x_1)_+, g_{k_1}(z), ...g_{k_l}(z)](H_K H_K^T)^{-1} H_K \boldsymbol{\theta} = \mathbb{E}_{\mathcal{D}_Y}[\hat{f}(z)]$ for fixed $K$.

**Lemma C.2.** *Let $K = \{k_1, ...k_l\}$ be a (possibly random) set of knot points in the data, and $Y$ be the vector of responses.*

*Let $F(K)$ be the subspace of vectors in $\mathbb{R}^n$ representable as evaluations piecewise-linear functions with optional discontinuities at points $K$. Let $S(K)$ be the subspace of vectors in $\mathbb{R}^n$ representable as evaluations (on the data) of linear splines with knots in $K$. Let $P_{(\cdot)}$ be the projection map. Now consider the following two estimators:*

1. *$\hat{f}_f$ is the fictitious estimator, whose predictions are given by:*

$$[\hat{f}_f(x_1)...\hat{f}_f(x_n)]^T = P_{F(K)}Y$$

2. $\hat{f}$ is AKORN's output, whose predictions are given by:

$$[\hat{f}(x_1)...\hat{f}(x_n)]^T = P_{S(K)}Y$$

*Then the following holds deterministically*

$$\sum_{i=1}^{n}(\hat{f}(x_i) - f(x_i))^2 \leq 2\sum_{i=1}^{n}(\hat{f}_f(x_i) - f(x_i))^2 + 2\sum_{i=1}^{n}(P_{S(K)}f(x_i) - f(x_i))^2$$

*Proof.* We abuse notation by letting $\hat{f} := [\hat{f}(x_1), ...\hat{f}(x_n)]$ and $\hat{f}_f = [\hat{f}_f(x_1), ...\hat{f}_f(x_n)]$ and suppressing the dependence of the spaces $S(K)$ and $F(K)$ on $K$. Now, with $\|\cdot\| = \|\cdot\|_2$, we obtain

$$\|\hat{f}_f - f\| \geq \|P_S\hat{f}_f - P_Sf\| = \|P_SP_FY - P_Sf\| =$$
$$\|P_SY - P_Sf\| = \|\hat{f} - P_Sf\|$$
$$\geq |\|\hat{f} - f\| - \|P_Sf - f\||$$

Where the first inequality is because projections are contractions, the second equality is because $S \subset F$. By the reverse triangle inequality, we obtain

Thus

$$\|\hat{f} - f\| \leq \|\hat{f}_f - f\| + \|P_Sf - f\|\|$$

$\square$

## C.3. Analysis for Piecewise Linear Estimates: $\|P_{F(K_r)}Y - \boldsymbol{\theta}\|_2^2$

For any random knot-set $K_r$, we now analyze the quality of the fit $P_{F(K_r)}Y$. This is necessary in two parts of our proof: for providing certificates $g$ and $h$ to be plugged into Lemma C.7 (with $K_r = K_f$ and $K_r = K_b$ respectively) and for bounding the error of the fictitious estimator in Lemma C.2 (with $K_r = K = K_f \cup K_b \cup \tilde{K}$).

The next Lemma gives a bias-variance decomposition of $\|P_{F(K_r)}Y - \boldsymbol{\theta}\|_2^2$.

**Lemma C.3.** *There exists an event $\mathcal{E}_0(\delta)$ which holds with probability at least $1 - \delta$ upon which the following holds for all knot-sets $K_r = \{k_1, ...k_l, k_{l+1} := n\}$ with their associated fit $\hat{f}_f = P_{F(K_r)}Y$*

$$\sum_{t=1}^{n}(\hat{f}_f(x_t) - f(x_t))^2 \lesssim \sum_{i=1}^{l}\left(\sum_{t=k_i}^{k_{i+1}-1}(f(x_t) - \mu_t)^2 + \sum_{t=k_i}^{k_{i+1}-1}\sigma_t^2 \log(\frac{n^3}{\delta})\right)$$

*where $\mu_t$ and $\sigma_t^2$ are the mean and variance of $\hat{f}_f(x_t)$ respectively, treating the knots as fixed.*

*Proof.* Begin by fixing a static interval $p_i = [k_i, k_{i+1} - 1]$. Further, fix some $t \in p_i$.

For any vector $q \in \mathbb{R}^n$, let $q^i = [q_{k_i}, ...q_{k_{i+1}-1}]^T$. Let $X_i$ collect only the covariates in $p_i$. Now note that

$$\hat{f}_f(x_t) = x_t^T(X_iX_i^T)^{-1}X_iY^i \sim \mathcal{N}(\mu_t, \sigma_t^2)$$

where $\mu_t = \langle x_t, (X_iX_i^T)^{-1}X_i\boldsymbol{\theta}^i\rangle$ and $\sigma_t^2 = \sigma^2 x_t^T(X_iX_i^T)^{-1}x_t$.

Thus, letting $W_t := \frac{\hat{f}_f(x_t) - \mu_t}{\sigma_t}$, we have that

$$\Pr[|W_t| \leq \sqrt{2\log(2n/\delta)}] > 1 - \delta/n$$

Or equivalently

$$\Pr[|\hat{f}_f(x_t) - \mu_t| \leq \sigma_t\sqrt{2\log(2n/\delta)}] > 1 - \delta/n \tag{10}$$

By reverse triangle inequality,

$$|\hat{f}_f(x_t) - \mu_t| = |\hat{f}_f(x_t) - \theta_t - (\mu_t - \theta_t)| > |\hat{f}_f(x_t) - \theta_t)| - |\mu_t - \theta_t|$$

So that Equation (10) implies

$$|\hat{f}(x_t) - \theta_t| \le |\mu_t - \theta_t| + \sigma_t \sqrt{2 \log(2n^3/\delta)}$$

holds with probability at least $1 - \delta/n^3$

Squaring each side and summing (and union bounding) over $t$ from $k_i$ to $k_{i+1} - 1$, this implies that with probability $1 - \delta/n^2$:

$$\sum_{t=k_i}^{k_{i+1}-1} (\hat{f}(x_t) - \theta_t)^2 \le 2 \sum_{t=k_i}^{k_{i+1}-1} (\mu_t - f(x_t))^2 + 4 \sum_{t=k_i}^{k_{i+1}-1} \sigma_t^2 \log(\frac{2n^3}{\delta})$$

where we have used $(a + b)^2 \le 2a^2 + 2b^2$.

To finish, cover all $n^2$ realizations of $p_i$. This means that with probability $1 - \delta$, *for all* $k_i, k_{i+1}$:

$$\sum_{t=k_i}^{k_{i+1}-1} (\hat{f}(x_t) - \theta_t)^2 \le 2 \sum_{t=k_i}^{k_{i+1}-1} (\mu_t - f(x_t))^2 + 4 \sum_{t=k_i}^{k_{i+1}-1} \sigma_t^2 \log(\frac{2n^3}{\delta})$$

In particular, we have the claimed decomposition for all $K_r$.

$\square$

We elaborate on the bias and variance terms in the following two lemmas.

**Lemma C.4.** *[Variance of $P_{F(K_r)}Y$] Within the setting of Lemma C.3, we have that for each $k_i \in K_r$*

$$\sum_{t=k_i}^{k_{i+1}-1} \sigma_t^2 \log(\frac{2T}{\delta})) \lesssim \sigma^2$$

*Proof.*

$$\sum_{t=k_i}^{k_{i+1}-1} \sigma_t^2 = \sigma^2 \sum_{t=k_i}^{k_{i+1}-1} x_t^T (X_i X_i^T)^{-1} x_t = 2\sigma^2$$

because the sum of the leverage scores is the number of parameters in the model. $\square$

For $K_r = K_0$ a knot-set from Lemma C.1 (on either a forward or backward pass), we have a control on the total bias of $P_{F(K_0)}Y$.

**Lemma C.5.** *[Bias] In the setting of Lemma C.1, we have the following on the good event $\mathcal{E}_1(\delta)$:*

$$\|P_{F(K_0)}\boldsymbol{\theta} - \boldsymbol{\theta}\|_2^2 = \tilde{O}(n^{1/5} C^{2/5})$$

*Proof.* Lemma C.1's success event guarantees that there is a vector $\boldsymbol{\eta} \in F(K_0)$ with

$$\|\boldsymbol{\eta} - \boldsymbol{\theta}\|_2^2 = \lesssim \sum_{i=1}^{l} 1 + n_i^{3/5} \|D^2 \boldsymbol{\theta^i}\|_2^{2/5}$$

$$\lesssim l + \sum_{i=1}^{l} n_i^{3/5} \|D^2 \boldsymbol{\theta^i}\|_2^{2/5}$$

Now using Holder's inequality with the dual norm pair $(5/3, 5/2)$, we obtain

$$\sum_{i=1}^{l} \|D^2\boldsymbol{\theta}^i\|_1^{2/5} n_i^{3/5} \leq (\sum_{i=1}^{l} \|D^2\boldsymbol{\theta}^i\|_1)^{2/5} (\sum_{i=1}^{l} n_i)^{3/5} \leq \|D^2\boldsymbol{\theta}\|_1^{2/5} n^{3/5} \leq C^{2/5} n^{1/5}$$

$\square$

An application of the previous three lemmas shows that $P_{F(K_0)}Y$ has small error for $K_0$ coming from the knot selection algorithm (i.e. both with $K_0 = K_f$ and $K_0 = K_b$).

**Corollary C.6.** *Let $\mathcal{E}_0(\delta/2)$ be the good event from Lemma C.3 and $\mathcal{E}_1(\delta/2)$ be the good event from Lemma C.1. Then on $\mathcal{E}_0(\delta/2) \cap \mathcal{E}_1(\delta/2)$ we have*

$$\|P_{F(K_0)}Y - \boldsymbol{\theta}\|_2^2 = \tilde{O}(n^{1/5}C^{2/5})$$

*Proof.* On $\mathcal{E}_0(\delta/2)$ we have

$$\|P_{F(K_0)}Y - \boldsymbol{\theta}\|_2^2 \leq \sum_{t=1}^{n} (\hat{f}_f(x_t) - f(x_t))^2 \lesssim \sum_{i=1}^{l} (\sum_{t=k_i}^{k_{i+1}-1} (f(x_t) - \mu_t)^2 + \sum_{t=k_i}^{k_{i+1}-1} \sigma_t^2 \log(\frac{n}{\delta}))$$

On the event $\mathcal{E}_1(\delta/2)$ from Lemma C.5 we can bound the bias term by $\tilde{O}(n^{1/5}C^{2/5})$. On this same event, we can bound the variance term by $\tilde{O}(\sigma^2 n^{1/5}C^{2/5}) = \tilde{O}(n^{1/5}C^{2/5})$ by Lemma C.4. $\square$

### C.4. Spline Existence

**Lemma C.7.** *Suppose $g \in F(K_f)$ and $h \in F(K_b)$ for $K_f \cap K_b = \{\}$. Let $K = K_f \cup K_b \cup \tilde{K}$ where $\tilde{K}$ contains all the crossover points between $g$ and $h$. Then there exists $f \in S(K)$ such that for all $x \in [0,1]$ there exists $\lambda_x \in [0,1]$ such that*

$$f(x) = \lambda_x g(x) + (1 - \lambda_x)h(x) \tag{11}$$

The idea behind the proof is simple. We construct a linear spline left-to-right that greedily sticks with whichever function of $g$ and $h$ is furthest from a change point, transitioning linearly between the two as necessary. We must include crossover points in order to ensure we do not exit the region between the curves $g$ and $h$ when we perform a "slide" from one to the other.

*Proof.* Assume $K = k_1, ...k_l$ is ordered. Let $k_{l+1} = 1$ for convenience.

We prove a stronger result, where we enforce the following additional requirements at each knot $k_i$.

1. $f(k_i) = g(k_i)$ or $f(k_i) = h(k_i)$

2. If $i \in [l-1]$ is such that $k_i \in \tilde{K}$ and $k_{i+1} \in K_f$, we have that $f(k_i) = h(k_i)$.

3. If $i \in [l-1]$ is such that $k_i \in \tilde{K}$ and $k_{i+1} \in K_b$, we have that $f(k_i) = g(k_i)$.

We construct $f$ in cases while iterating over knots.

**Base case:** If $k_1 \in K_f$, let $f(z) = h(z)$ for $z \in [0, k_1]$. If $k_1 \in K_b$, let $f(z) = g(z)$ for $z \in [0, k_1]$. If $k_1 \in \tilde{K}$, then let $f(z) = g(z)$ for $z \in [0, k_1]$ if $k_2 \in K_b$ and $f(z) = h(z)$ for $z \in [0, k_1]$ if $k_2 \in K_f$.

**"Inductive" step:** Assume that $(f : [0, k_{i-1}] \to \mathbb{R}) \in S([k_1, ...k_{i-1}])$ is constructed such that the above requirements are satisfied for all knots $k_1, ...k_{i-1}$. We now extend $f$ to $(f : [0, k_i] \to \mathbb{R}) \in S([k_1, ...k_i])$.

**Case 0:** $f(k_{i-1}) = g(k_{i-1})$ **and $k_{i-1}$ is the last knot ($i - 1 = l$)**

We can extend $f$ to $[0, k_{l+1}] = [0, 1]$ by letting $f(z) = g(z)$ for $z \in [k_{i-1}, k_i]$. Because $g$ is linear on $[k_{i-1}, k_i]$, $f$ is in $S(K)$. Equation 11 holds by construction, and requirements 1, 2 and 3 all hold by our iterative hypothesis.

**Case 1:** $f(k_{i-1}) = g(k_{i-1})$ **and $k_i \in K_b$**

Extend $f$ by letting $f(z) = g(z)$ for all $z \in [k_{i-1}, k_i]$. Because $g$ is linear on $[k_{i-1}, k_i]$ and $g(k_{i-1}) = f(k_{i-1})$, we still have that $f \in S([k_1, ...k_i])$. We also have $f(z) = g(z)$ for all $z \in [k_{i-1}, k_i]$ and $f(k_i) = g(k_i)$.

**Case 2:** $f(k_{i-1}) = g(k_{i-1})$ **and** $k_i \in K_f$

Extend $f$ by letting $f(z) = g(z) + (h(z) - g(z))\frac{z - k_{i-1}}{k_i - k_{i-1}}$ for $z \in [k_{i-1}, k_i]$. By construction, we are also assured that $k_{i-1} \notin \tilde{K}$ (because otherwise we would have $f(k_{i-1}) = h(k_{i-1})$). Because $g$ and $h$ do not cross on the interval $[k_{i-1}, k_i]$, we have that $f$ is in $S([k_1, k_i])$ and satisfies Equation 11 (with $x$ restricted to $[0, k_i] \cup [k_{i-1}, k_i]$). By construction, we also have that $f(k_i) = h(k_i)$ (satisfying requirement 1). By our inductive hypothesis, the extended version of $f$ satisfies requirements 2 and 3.

**Case 3:** $f(k_{i-1}) = g(k_{i-1})$ **and** $k_i \in \tilde{K}$ **and** $k_{i+1} \in K_b \cup \{k_{l+1}\}$

Define $f(z) = g(z)$ for all $z \in [k_{i-1}, k_i]$. Because $g$ is linear on $[k_{i-1}, k]$, we still have that $f \in S([k_1, ...k_i])$. We also have $f(z) = g(z)$ for all $z \in [k_{i-1}, k_i]$ and $f(k_i) = g(k_i) = h(k_i)$.

Further, by construction we have that requirements 1 and 3 hold after extension. By hypothesis, requirement 2 still holds.

**Case 4:** $f(k_{i-1}) = g(k_{i-1})$ **and** $k_i \in \tilde{K}$ **and** $k_{i+1} \in K_f$

Define $f(z) = g(z) + (h(z) - g(z))\frac{z - k_{i-1}}{k_i - k_{i-1}}$ for $z \in [k_{i-1}, k_i]$. Because $k_i \in \tilde{K}$, we know that $g$ and $h$ do not cross on the interval $[k_{i-1}, k_i]$. Because $g$ and $h$ do not cross on the interval $[k_{i-1}, k_i]$, we have that $f$ satisfies Equation 11 (with $x$ restricted to $[0, k_i] \cup [k_{i-1}, k_i]$).

Furthermore, by construction, requirements 1 and 2 hold after extension. By hypothesis, requirement 3 still holds.

The remaining cases are symmetric to the above ones (i.e. the orders of $g$ and $h$ and $K_f$ and $K_b$ are flipped). We can iterate this scheme left-to-right over all knots $k \in K$ to prove the result. $\qquad\square$

Generally speaking, the odds of $K_f$ and $K_b$ sharing knots when generated according to AKORN are not high. However, the following corollary shows that we can handle this case should it occur.

**Corollary C.8.** *Suppose $g \in F(K_f)$ and $h \in F(K_b)$, with $K_f \cap K_b \neq \{\}$. Let $K = K_f \cup K_b \cup \tilde{K} \cup Q$, where $Q = \{x_{i-1} : x_i \in K_b \cap K_f\}$ and $\tilde{K}$ is the set of crossover points of $g$ and $h$. Then there exists $s \in S(K)$ such that for all $x \in [0, 1]$ there exists $\lambda_x \in [0, 1]$ such that*

$$s(x) = \lambda_x g(x) + (1 - \lambda_x)h(x) \tag{12}$$

*Proof.* Let $Q = x_{q_1}, ...x_{q_w}$ be ordered. On each interval $[x_{q_i}, x_{q_{i+1}-1}]$ we may construct a corresponding linear spline $s_i$ using Lemma C.7. We can then construct $s$ by linearly interpolating between the various $s_i$ using the knots in $Q$. $\qquad\square$

### C.5. Proof of Theorem 6.2

**Theorem C.9** (Theorem 6.2)**.**

*Proof.* Let $\hat{f} = P_{S(K)}f$. Let $\mathcal{E}_0(\delta/2)$ be the event from Lemma C.3. On $\mathcal{E}_0(\delta/2)$, the following holds for all knot-sets $K = \{k_1, ...k_l\}$, $l > 0$ with $P[\mathcal{E}_0] \geq 1 - \delta/2$.

$$\sum_{t=1}^{n}(\hat{f}(x_t) - f(x_t))^2 \leq \sum_{t=1}^{n}(\hat{f}_f(x_t) - f(x_t))^2 + \sum_{t=1}^{n}(P_{S(K)}f(x_t) - f(x_t))^2$$

$$\leq \sum_{i=1}^{l}\sum_{t=k_i}^{k_{i+1}-1}(f(x_t) - \mathbb{E}[\hat{f}_f(x_t)])^2 + \sum_{i=1}^{l}\sum_{t=k_i}^{k_{i+1}-1}\sigma_t^2\iota(\delta) + \sum_{t=1}^{n}(P_{S(K)}f(x_t) - f(x_t))^2$$

$$\leq \sum_{i=1}^{l}\sum_{t=k_i}^{k_{i+1}-1}(f(x_t) - \mathbb{E}[\hat{f}_f(x_t)])^2 + 2\sigma^2 l + \sum_{t=1}^{n}(P_{S(K)}f(x_t) - f(x_t))^2 \tag{13}$$

where the first line holds deterministically by Lemma C.2, the second line holds on the event $\mathcal{E}_0$ from Lemma C.3, and the third line holds by Corollary C.4.

Now, let $K_f$ and $K_b$ be the random knot-sets from isotonic and reverse isotonic runs of Algorithm 1 and let $K = K_f \cup K_b \cup \tilde{K}$. Let $\mathcal{E}_1^f(\delta/4)$ be the event from Lemma C.1 applied to the forward run of AKORN. By the conclusion of Lemma C.1, we have $|K_f| = \tilde{O}(n^{1/5}C^{2/5})$. By corollary C.5, we also have $\|f - P_{F(K_f)}f\|_2^2 \leq \tilde{O}(n^{1/5}C^{2/5})$ on $\mathcal{E}_1^f(\delta/4)$. Similarly, we let $\mathcal{E}_1^b(\delta/4)$ be the event from Lemma C.1 applied to the backward run of AKORN, upon which we have $|K_b| = \tilde{O}(n^{1/5}C^{2/5})$ on $\mathcal{E}_1^b(\delta/4)$ and $\|f - P_{F(K_b)}f\|_2^2 \leq \tilde{O}(n^{1/5}C^{2/5})$.

Let $\mathcal{E}_1 = \mathcal{E}_1^f \cap \mathcal{E}_1^b$. By union bound, $\Pr[\mathcal{E}_1] \geq 1 - \delta/2$. On $\mathcal{E}_1$, we may bound the first term in Equation 13 as

$$\sum_{i=1}^{l} \sum_{t=k_i}^{k_{i+1}-1} (f(x_t) - \mathbb{E}[\hat{f}_f(x_t)])^2 = \|f - P_{F(K)}f\|_2^2 \leq \|f - P_{F(K_f)}f\|_2^2 = \tilde{O}(n^{1/5}C^{2/5})$$

because $F(K_f) \subset F(K)$

and the second term as $2\sigma^2|K| \leq 2\sigma^2(4 \times |K|) = \tilde{O}(n^{1/5}C^{2/5})$.

All that remains is to bound the final term of Equation 13. To do this, first define $\mathcal{E}_3 = \mathcal{E}_0 \cap \mathcal{E}_1$. By construction, all the previous bounds still hold on $\mathcal{E}_3$, and we have $\Pr[\mathcal{E}_3] > 1 - \delta$ (by a union bound). Now apply Lemma C.7 to $g = P_{F(K_f)}Y$ and $h = P_{F(K_b)}Y$ to get a function $s \in S(K)$ that lies in between $g$ and $h$ for all $x \in [0,1]$ (i.e. $s(x_i) = \lambda_{x_i}g(x_i) + (1 - \lambda_{x_i})h(x_i)$ for $\lambda_{x_i} \in [0,1]$). Using convexity of square loss and the bounds on the error of $g$ and $h$ from the previous paragraph, we have

$$\|P_{S(K)}f - f\|_2^2 \leq \|s - f\|_2^2 = \sum_{i=1}^{n} (\lambda_{x_i}g(x_i) + (1 - \lambda_{x_i})h(x_i) - f(x_i))^2$$

$$\leq \sum_{i=1}^{n} \lambda_{x_i}(g(x_i) - f(x_i))^2 + \sum_{i=1}^{n} (1 - \lambda_{x_i})(h(x_i) - f(x_i))^2 = \tilde{O}(n^{1/5}C^{2/5})$$

Where the inequality holds by convexity of $\ell^2$-loss and the final bound holds on $\mathcal{E}_3$ due to Lemma C.6, which guarantees small error for $g$ and $h$ on $\mathcal{E}_0 \cap \mathcal{E}_1^f \subset \mathcal{E}_3$ and $\mathcal{E}_0 \cap \mathcal{E}_1^b \subset \mathcal{E}_3$ respectively.

$\square$

### C.6. Helper Lemmas

**Lemma C.10.** *[Simplification of Lemma 4 from (Rhee & Talagrand, 1986)] Let $Z \sim \mathcal{N}(0, \Sigma)$. Then*

$$\Pr[\|Z\| \geq t] \leq \exp \frac{-t^2}{2\mathrm{tr}(\Sigma)}$$

$\square$

## D. Proofs for ADDLE

The proofs in this section represent fairly straightforward extensions of those found in (Baby et al., 2021).

We will prove the following theorem. If $N = n$, then Theorem D.1 becomes Theorem 6.1.

**Theorem D.1.** *Consider equally spaced design points, $\{x_t = t/n\}_{t=p}^{p+N}$, for $p \geq 1$ and $p + N \leq n$. Let $C := TV_1[f|_{[x_p, x_{p+N}]}]$. Let $\{\hat{y}_t\}_{t=1}^{n}$ be the the predictions generated by Algorithm 5 when fed these data in order. With probability $1 - \delta$, the total squared error satisfies:*

$$\sum_{t=1}^{n} (\hat{y}_t - f(x_t))^2 = \tilde{O}(N^{\frac{3}{5}}C^{2/5}/n^{2/5})$$

*where $\tilde{O}$ hides constants (including $\sigma$) and polylog factors of $n$ and $\delta$.*

*Proof Sketch*: The proof of the bound for ADDLE in Theorem 6.1 follows along the same lines as the proof of Aligator in (Baby et al., 2021). The idea is that, for any $f$, there exists a not-too-large partition of $\{x_1, ...x_n\}$ into intervals such that $f$ is approximately linear within each interval (as measured by $TV_1$). On each of these intervals, the linear expert who starts at the beginning of the interval achieves low error. Furthermore, by the adaptivity property of FLH (Proposition D.2 below), ADDLE competes with the best expert on each interval. Thus, summing over intervals, we observe that ADDLE achieves the optimal rate.

*Beginning of formal proof*

The main tool is the following lemma, which states that FLH competes with each expert in each interval.

**Proposition D.2** ((Hazan & Seshadhri, 2007)). *Suppose the loss functions are exp-concave with parameter $\alpha$. For any interval $I = [r, s]$ in time, the algorithm FLH Fig.4 with learning rate $\zeta = \alpha$ gives $O(\alpha^{-1}(\log r + \log |I|))$ regret against the base learner in hindsight.*

$\square$

The following lemma follows instantly from a subgaussian tail bound and a union bound.

**Lemma D.3** (Lemma 16 from (Baby et al., 2021)). *Let $\mathcal{V}$ be the event that $|\epsilon_t| \leq \sigma\sqrt{2\log 4n/\delta}$. Then $\Pr[\mathcal{V}] \geq 1 - \delta/2$*

$\square$

Note that, conditioned on $\mathcal{V}$, the quantity $B$ from Algorithm 5 upper bounds $\theta_t$ for all $t$.

Define the filtration $\mathcal{F}_j = \sigma\{y_1, ...y_{j-1}\}$ and let $\mathbb{E}_j[\cdot] = \mathbb{E}[\cdot|\mathcal{F}_j]$ and $\mathrm{Var}_j[\cdot] = \mathrm{Var}[\cdot|\mathcal{F}_j]$.

Let $\hat{y}_t$ be ADDLE's prediction at time $t$. Let $\hat{z}_t^r$ be the prediction at time $t$ of the linear expert that starts at time $r$. Let $R_\sigma = 16(1 + \sigma\sqrt{\log 4n/\delta})^2$. Let $\tilde{\sigma} = \max\{\sigma\sqrt{\log 4n/\delta}, 1\}$.

The following Lemma ensures that $B$, as defined in Algorithm 3 is an upper-bound on $f$ .

**Lemma D.4.** *Let $B = \max_i |y_i| + \tilde{\sigma}$. On the event $\mathcal{V}$, we have that $|f(x_i)| \leq B \leq 1 + 2\tilde{\sigma}$ for every $i \in [n]$*

*Proof.* On $\mathcal{V}$, we have that, for any $i$, $y_i \in (f(x_i) - \tilde{\sigma}, f(x_i) + |\tilde{\sigma})$. Therefore, $B \geq |y_i| + \tilde{\sigma} \geq |f(x_i)|$. Also, $B \leq \max_i |y_i| + \tilde{\sigma} \leq 1 + 2\tilde{\sigma}$. $\square$

**Lemma D.5.** *Let $I = [r, s]$ be any interval. On the event $\mathcal{V}$, the predictions $\hat{y}_j$ made by ADDLE satisfy:*

$$\sum_{j=r}^{s} (\hat{y}_j - y_j)^2 \leq \sum_{j=r}^{s} (\hat{z}_j^r - y_j)^2 + \frac{2\log n}{R_\sigma}$$

*Proof.* On the event $\mathcal{V}$, each loss function $(\cdot - y_t)^2$ is $R_\sigma^{-1} := \eta$ exp-concave.

Now apply Lemma D.2 and bound $r, s - r \leq n$. $\square$

The following Lemma is proved as Lemma 18 in (Baby et al., 2021), recalling that $|\hat{z}_j^r - \theta_j| \leq 2(1 + \tilde{\sigma})$.

**Lemma D.6.** *For any $j \in [n]$, we have*

1. $\mathbb{E}_j[(y_j - \hat{z}_j^r)^2 - (y_j - \theta_j)^2|\mathcal{V}] = \mathbb{E}_j[(\hat{z}_j^r - \theta_j)^2|\mathcal{V}]$.

2. $\mathrm{Var}_j[(y_j - \hat{z}_j^r(j))^2 - (y_j - \theta_j)^2|\mathcal{V}] \leq R_\sigma \mathbb{E}_j[(\hat{z}_j^r(j) - \theta_j)^2|\mathcal{V}]$.

**Lemma D.7.** *(Freedman type inequality, (Beygelzimer et al., 2011)) For any real valued martingale difference sequence $\{Z_t\}_{t=1}^{T}$ with $|Z_t| \leq R$ it holds that,*

$$\sum_{t=1}^{T} Z_t \leq \eta(e - 2) \sum_{t=1}^{T} \mathrm{Var}_t[Z_t] + \frac{R\log(1/\delta)}{\eta}, \tag{14}$$

*with probability atleast $1 - \delta$ for all $\eta \in [0, 1/R]$.*

We use these Lemmas to define and bound two Martingale Difference Sequences (MDS). Again, compare to Lemma 19 in (Baby et al., 2021).

**Lemma D.8.** *Condition on $\mathcal{V}$. For any interval $[r, s]$, it holds with probability at least $1 - \delta$ that*

1. $\sum_{j=r}^{s}(y_j - \hat{z}_j^r)^2 - (y_j - \theta_j)^2 \leq (e-1)\sum_{j=r}^{s}(\hat{z}_j^r - \theta_j)^2 + R_\sigma^2 \log 4/\delta,$

2. $\sum_{j=r}^{s}(y_j - \hat{y}_j)^2 - (y_j - \theta_j)^2 \geq (3-e)\sum_{j=r}^{s}(\hat{y}_j - \theta_j)^2 - R_\sigma^2 \log 4/\delta.$

*Proof.* We continue to condition on $\mathcal{V}$. By Lemma D.6, $Z_j := (\hat{z}_j^r - y_j)^2 - (y_j - \theta_j)^2 - (\hat{z}_j^r - \theta_j)^2 = 2\epsilon_j(\hat{z}_j^r - \theta_j)$ is an MDS. Note that, because of the truncation step, $|Z_j| = 2|(\hat{z}_j^r - \theta_t)(\theta_t - y_t)| \leq 2(2B)\tilde{\sigma} \leq 4(1 + 2\tilde{\sigma})\tilde{\sigma} \leq R_\sigma$

By Lemma D.7 with $\eta = \frac{1}{R_\sigma}$, we therefore obtain

$$\sum_{j=r}^{s}(\hat{z}_j^r - y_j)^2 - (y_j - \theta_j)^2 - (\hat{z}_j^r - \theta_j)^2 \leq (e-2)\sum_{j=r}^{s}(\hat{z}_j^r - y_j)^2 + R_\sigma^2 \log\frac{1}{\delta}$$

with probability $1 - \delta$.

We obtain the second inequality by an identical argument with the MDS

$$\sum_{j=r}^{s}(\hat{y}_j - \theta_j)^2 + (y_j - \theta_j)^2 - (\hat{y}_j - y_j)^2$$

Union bounding over 1 and 2 gives the result. □

Now, note that by Lemma D.5, we have that

$$\sum_{j=r}^{s}(\hat{y}_j - y_j)^2 - (y_j - \theta_j)^2 \leq \sum_{j=r}^{s}(\hat{z}_j^r - y_j)^2 - (y_j - \theta_j)^2 + \frac{2\log n}{R_\sigma}$$

So that, by Lemma D.8

$$(3-e)\sum_{j=r}^{s}(\hat{y}_j - \theta_j)^2 - R_\sigma^2 \log\frac{4}{\delta} \leq \sum_{j=r}^{s}(\hat{y}_j - y_j)^2 - (y_j - \theta_j)^2 \leq \sum_{j=r}^{s}(\hat{z}_j^r - \theta_j)^2 + R_\sigma^2 \log\frac{4}{\delta} + \frac{2\log n}{R_\sigma}$$

which (for fixed $r, s$) leads to the following high-probability relation:

$$\sum_{j=r}^{s}(\hat{y}_j - \theta_j)^2 \leq \frac{(e-1)}{(3-e)}\sum_{j=r}^{s}(\hat{z}_j^r - \theta_j)^2 + 2R_\sigma^2 \log 4/\delta + 2\log n/R_\sigma$$

Let's union bound over the $n^2$ possibilities for $r, s$ to get that, with probability $1 - \delta$ for all intervals $[r, s]$

$$\sum_{j=r}^{s}(\hat{y}_j - \theta_j)^2 \leq \frac{(e-1)}{(3-e)}\sum_{j=r}^{s}(\hat{z}_j^r - \theta_j)^2 + 2R_\sigma^2 \log 4n^2/\delta + 2\log n/R_\sigma \tag{15}$$

Since we're conditioning on $\mathcal{V}$, observe that, if $\hat{w}_j^r = \texttt{Predict}(\{x_t, y_t\}_{t=r}^{j-1}, x_j)$ is the prediction from a hypothetical unbounded linear expert (Algorithm 4), $(\hat{z}_j^r - \theta_j)^2 \leq (\hat{w}_j^r - \theta_j)^2$. Thus, from this point forward, we consider $\hat{w}_j^r$ instead of $\hat{z}_j^r$.

Now, noting that (for $j > r + 1$) $\hat{w}_j^r \sim \mathcal{N}(w_j^r, \underbrace{\sigma^2 x_j^T (X_{j-1}X_{j-1}^T)x_j}_{\sigma_j^2})$, we have by direct computation (Lemma E.2 in

Appendix E)

$$\sum_{j=r}^{s}(\hat{w}_j^r - \theta_j)^2 \le 3\sigma^2 \log 2n/\delta \log en + \sum_{j=r+2}^{s}(w_j^r - \theta_j)^2$$

Plugging this result into Equation 15, we summarize in the following lemma.

**Lemma D.9.** *Condition on $\mathcal{V}$. Within this conditioning, With probability $1 - \delta/2$, the following bound holds over all intervals $[r, s]$*

$$\sum_{j=r}^{s}(\hat{y}_j - \theta_j)^2 \le \tilde{O}(1 + \sum_{j=r+2}^{s}(w_j^r - \theta_j)^2 + \sigma_j^2)$$

*where $\tilde{O}(\cdot)$ hides only constants, as well as log factors of $n$ and $\delta$, and where the sum is considered to be zero if $s \le r + 1$.*

We can also now uncondition on $\mathcal{V}$, and union bound over $\mathcal{V}^c$ and $\mathcal{V} \cap \mathcal{C}^c$, where $\mathcal{C}$ is the good event from Lemma D.9

**Lemma D.10.** *With probability $1 - \delta$, the following bound holds over all intervals $[r, s]$*

$$\sum_{j=r}^{s}(\hat{y}_j - \theta_j)^2 \le \tilde{O}(1 + \sum_{j=r+2}^{s}(w_j^r - \theta_j)^2 + \sigma_j^2)$$

*where $\tilde{O}(\cdot)$ hides only constants, as well as log factors of $n$ and $\delta$, and where the sum is considered to be zero if $s \le r + 1$.*

As compute in Appendix E (Equation 16 of Lemma E.2), $\sum_{j=r}^{s}\sigma_j^2 = \tilde{O}(1)$. By Lemma E.1 we obtain $\sum_{j=r+2}^{s}(w_j^r - \theta_j)^2 = TV_1(\theta[r:s])^2|r - s|^3/n^2$ for equally spaced $\{x_j = j/n\}_{j=r}^{s}$. This leads to the follow lemma.

**Lemma D.11.** *Let $\mathcal{P} = [r_1 = 1, r_2] \cup \{[r_i, r_{i+1} - 1]\}_{i=2}^{l-2} \cup [r_{l-1}, r_l - 1]$ be any partition of $[n]$ into contiguous intervals with $r_l = n + 1$. Let $n_i = r_{i+1} - r_i$ be the length of the $i$th interval, and $TV_1(i) := TV_1(\theta[r_i : r_{i+1} - 1])$. Then, with probability $1 - \delta$:*

$$\sum_{j=1}^{n}(\hat{y}_j - \theta_j)^2 \le \tilde{O}(\sum_{i=1}^{l-1}(TV_1(i)^2 n_i^3/n^2 + 1))$$

Now consider the partitioning scheme that scans left to right from $p$ to $p + N$, and adds points to the current bin so long as $TV_1(\theta[\text{current bin}]) < \frac{n}{\text{current bin size}^{3/2}}$. It follows immediately that, the $TV_1$ inside each bin satisfies $TV_1(\text{bin})^2 \le n^2/(\text{bin size})^3$. This is analagous to the $TV_0$ case from (Baby et al., 2021). As can be seen in Lemma 23 in (Baby & Wang, 2020), the total number of bins in this partition is bounded by $O(N^{3/5}C^{2/5}/n^{2/5})$. Thus, letting $\mathcal{P}$ be this partition, Lemma D.11 becomes Theorem D.1.

## E. Some missing computations

**Lemma E.1** (Bias of linear regression). *Suppose $x_1, ...x_n$ are sorted covariates such that $\max_{j=2,...n}(x_j - x_{j-1}) \le \log n/(p_0 n)$ for some constant $p_0 > 0$. Let $\theta_j := f(x_j)$, so that our data is $\{(x_i, \theta_i)\}_{i=1}^{n}$. Further, consider some subset $\{(x_i, \theta_i)\}_{i=r}^{N}$. Let $\hat{l}(z) = \hat{a} + \hat{b}x$ be the linear least squares fit trained on this subset. Then the error of $\hat{l}$ is bounded as*

$$\sum_{i=r}^{N}(\hat{l}(x_i) - \theta_i)^2 \le N^3 TV_1(\theta[r:N])^2 \log^2 n/(p_0^2 n^2)$$

*In the special case where $x_i = i/n$ for $i = 1, ..., n$, we have*

$$\sum_{i=r}^{N}(\hat{l}(x_i) - \theta_i)^2 \le N^3 TV_1(\theta[r:N])^2/n^2$$

*Proof.* WLOG suppose $r = 1$.

Define $\bar{a}$ to be equal to $\theta_1$ and $\bar{b}$ to be $\frac{1}{N}\sum_{j=1}^{N} s_j$, where for $j > 1$ we let $s_j = \frac{\theta_j - \theta_{j-1}}{x_j - x_{j-1}}$ be the slope from the datapoint $j - 1$ to $j$. We then have

$$\sum_{i=1}^{N}(\hat{a} + \hat{b}x_i - \theta_i)^2$$

$$\overset{(1)}{\leq} \sum_{i=1}^{N}(\bar{a} + \bar{b}(x_i - x_1) - \theta_1 - \sum_{k=2}^{i} s_k(x_k - x_{k-1}))^2$$

$$= \sum_{i=1}^{N}(\bar{b}\sum_{k=2}^{i}(x_k - x_{k-1}) - \sum_{k=2}^{i}(x_k - x_{k-1})s_k)^2$$

$$= \sum_{i=1}^{N}(\sum_{k=2}^{i}(\bar{b} - s_k)(x_k - x_{k-1}))^2$$

$$\leq \sum_{i=1}^{N}\sum_{k=2}^{i}(\bar{b} - s_k)^2 \sum_{k=2}^{i}(x_k - x_{k-1})^2$$

$$\overset{(2)}{\leq} \sum_{i=1}^{N} NTV_1(\theta[1:N])^2 \times \frac{N \log^2 n}{p_0^2 n^2}$$

$$\leq N^3 TV_1(\theta[1:N])^2 \times \log n^2/(p_0^2 n^2)$$

(1) holds because $\hat{a}, \hat{b}$ minimize square loss among linear functions. (2) holds because for any vector $z \in \mathbb{R}^d$, we have $\sum_{i=1}^{d}(z[i] - \bar{z})^2 \leq dTV_0(z)^2$, where $\bar{z} = \sum_{i=1}^{d} z_i/d$. This fact is applied with $z_j = s_j$ for $j = 1, ...n$, which leads to $TV_0(z) = TV_1(\theta[1:N])$.

The equal-spacing case follows from an identical argument where $(x_k - x_{k-1})^2$ is instead set to $\frac{1}{n^2}$ □

**Lemma E.2** (Running variance for ADDLE). *Consider a set of covariates, $\{x_t = t/n\}_{t=1}^{n}$, and responses $\{y_t = f(x_t) + \epsilon_t\}_{t=1}^{n}$. For any interval $[a, b] \subset [1, n]$ with length $l > 2$, consider $\hat{z}_t$ to be the prediction of online linear regression (Algorithm 4) at time $t$ after starting at time $a$. Let $z_t = \mathbb{E}[\hat{z}_t]$. Then with probability $1 - \delta$:*

$$\sum_{t=a}^{b}(\hat{z}_t - z_t)^2 \leq 2\sigma^2 \log\left(\frac{2n}{\delta}\right) \log en + \sigma^2 \log 2/\delta$$

*Proof.* Without loss of generality let $[a, b] = [1, l]$. Start by fixing $t \in [3, l-1]$. Let $X_t \in \mathbb{R}^{2 \times t}$ have columns $\{[x_i, 1]^T\}_{i=1}^{t}$.

$$\hat{z}_{t+1} - z_{t+1} = x_{t+1}^T(X_t X_t^T)^{-1}X_t(Y - \boldsymbol{\theta}) \sim \mathcal{N}(0, \sigma^2 x_{t+1}^T(X_t X_t^T)^{-1}x_{t+1})$$

Letting $\sigma_t^2 = \sigma^2 x_{t+1}^T(X_t X_t^T)^{-1}x_{t+1}$, and applying a gaussian tail bound, we obtain:

$$\Pr[|\hat{z}_{t+1} - z_{t+1}| \leq \sigma_t \sqrt{2 \log 2n/\delta}] \geq 1 - \delta/n$$

So that

Squaring each side, then union bounding over $t \in [3, l]$ and summing up, we have that, with probability $1 - \delta$:

$$\sum_{t=3}^{l}(\hat{z}_{t+1} - z_{t+1})^2 \leq 2\sigma^2 \log(2n/\delta) \sum_{t=3}^{l} x_t^T(X_t X_t^T)^{-1}x_t \tag{16}$$

Thus, we need to analyze the "out-of-sample leverage scores". Observe that $(X_t X_t^T)^{-1}$ is given by:

$$(X_t X_t^T)^{-1} = \frac{1}{t^2 \times \frac{1}{t} \times \sum_i (x_i - \overline{x})^2} \begin{bmatrix} t & -\sum_{i=1}^t x_i \\ -\sum_{i=1}^t x_i & \sum_{i=1}^t x_i^2 \end{bmatrix}$$

If we assume equally spaced points with pairwise distance $1/n$, we can compute

$$\frac{n^2}{t^2(t+1)(t-1)} \begin{bmatrix} t & -t(t+1)/2n \\ -t(t+1)/2n & t(t+1)(2t+1)/6n^2 \end{bmatrix} = \begin{bmatrix} n^2/t(t+1)(t-1) & -n/2t(t-1) \\ -n/2t(t-1) & (2t+1)/6t(t-1) \end{bmatrix}$$

So that

$$x_{t+1}^T Z_t x_{t+1} = \frac{(t+1)^2}{n^2} \times \frac{n^2}{t(t+1)(t-1)} - 2 \times \frac{(t+1)}{2t(t-1)} + \frac{2t+1}{6(t-1)t} = \frac{2t+1}{6t-1} \times \frac{1}{t} \le \frac{1}{t}$$

Thus, using $\sum_{t=1}^l \frac{1}{t} \le \log n + 1 = \log(en)$ when we plug in to Equation 16, we are left with:

$$\sum_{t=3}^l (\hat{z}_{t+1} - z_{t+1})^2 \le 2\sigma^2 \log(2n/\delta) \log en \tag{17}$$

To finish, recall that for $t = 1$, online regression predicts $0$ deterministically, so that $\hat{z}_t - z_t = 0$. For $t = 2$, it predicts $y_1$, which will yield a $\mathcal{N}(0, \sigma^2)$ summand, which can be bounded with high probability. We tack these terms onto the above display after a union bound.

$\square$

# F. Uneven and Random Covariates

## F.1. Theorem statements for uneven covariates

In this section, we explain how ADDLE can be generalized to handle the case of uneven covariates. These proofs rely on a minor algorithmic change: We replace each clipped online linear regression expert of Figure 4 by a clipped Vovk-Azoury-Warmuth (VAW) forcaster with the same start-point (see (Baby & Wang, 2020; Cesa-Bianchi & Lugosi, 2006) for descriptions of the VAW forcaster).

We show how to prove the following generalization of Theorems 6.1

**Theorem F.1.** *For some $p_0 > 0$, consider sorted design points $0 \le x_1, ...x_n \le 1$ such that $\max_{j=2,...n} |x_j - x_{j-1}| \le \frac{\log n}{p_0 n}$. Let $f$ be a function with $C := TV_1[f, \mathcal{D}_X]$, and consider responses $\{y_t\}$ coming from the regression model. Let $\{\hat{y}_t\}_{t=1}^n$ be the the predictions generated by ADDLE, now with (clipped) VAW forcasters as experts. With probability $1 - \delta$, the total squared error satisfies:*

$$\sum_{t=1}^n (\hat{y}_t - f(x_t))^2 = \tilde{O}(n^{1/5} C^{2/5})$$

*where $\tilde{O}$ hides constants (including $\sigma$) and polylog factors of $n$ and $\delta$.*

This leads directly to the corresponding generalization of Theorem 6.2.

**Theorem F.2.** *For some $p_0 > 0$, consider sorted design points $0 \le x_1, ...x_n \le 1$ such that $\max_{j=2,...n} |x_j - x_{j-1}| \le \frac{\log n}{p_0 n}$. Let $f$ be a function with $C := TV_1[f, \mathcal{D}_X]$, and consider responses $\{y_t\}$ coming from the regression model. Let $\hat{f}$ be the function returned by AKORN. Then, with probability $1 - \delta$, the average square error satisfies:*

$$\frac{1}{n} \sum_{t=1}^n (\hat{f}(x_t) - f(x_t))^2 = \tilde{O}(n^{-4/5} C^{2/5})$$

*where $\tilde{O}$ hides constants (including $\sigma$) and polylog factors of $n$ and $\delta$.*

### F.2. Proof steps for Theorem's F.1 and F.2

**Steps for Theorem F.1**

We now consider as our experts clipped linear Vovk-Azoury-Warmuth (VAW) forecasters ((Cesa-Bianchi & Lugosi, 2006)) starting at time $r$ for each $r \in [n]$. This is a very minor change from the original linear regression experts, and does not affect computational or statistical efficiency. The VAW expert starting at $r$ is fed data $D_{r,s} := \{(x_j, y_j)\}_{j=r}^s$ in an online fashion, and produces estimates $\hat{w}_r^r, ... \hat{w}_s^r$.

Notice that, even with these changes to our setting, we can run the proof of Appendix D up until Equation (15), where now $\hat{z}_j^r$ is the clipped VAW expert that starts at time $r$. We can still replace this expert with a hypothetical unclipped expert, $\hat{w}_j^r$.

By Lemma 24 of (Baby & Wang, 2020), we have:

$$\sum_{j=r}^s (\theta_j - \hat{w}_j^r)^2 \leq \sum_{j=r}^s (\theta_j - l_{r:s}(x_j))^2 + \|u\|_2^2 + \tilde{O}(1)$$

where $l_{r:s}(x_i) = u^T x_i$ is the offline linear least squares estimate trained on noiseless data $(x_j, \theta_j)$ $j = r, ...s$. From Corollary 40 of (Baby & Wang, 2020), we have $\|u\|_2^2 = O(1)$. By Lemma E.1 we have that the first term is bounded by $|r - s|^3 TV_1(\theta[r:s])^2/n^2$.

Plugging this argument into Equation (15), we recover Lemma D.11's statement that, with high probability, for any partition $\mathcal{P} = p_1, ...p_l$ with $p_i = [x_{r_i}, x_{r_{i+1}-1}]$

$$\sum_{j=1}^n (\hat{y}_j - \theta_j)^2 \leq \tilde{O}(\sum_{i=1}^l (TV_1(i)^2 n_i^3/n^2 + 1))$$

To complete the proof, we may now construct the oracle partition in the same way as before, where $TV_1$ of a bin is computed with respect to realized covariate spacing.

**Steps for Theorem F.2**

All of the spline approximation results of Appendix C go through without technical changes. Now that ADDLE has been generalized to the uneven covariate setting, Lemma C.1. also goes through by an application of Lemma E.1 to the bias of the linear fits $\hat{a}_t$ (concentration is not an issue, as we still have $\sum_{j=1}^n x_j^T (XX^T)^{-1} x_j = 2\sigma^2$).

### F.3. Theorem for random covariates

First, we cite a result that tells us that draws are roughly evenly spaced when they come from a distribution whose density is bounded below on $[0, 1]$. We do not have control over the probability of the good event in this lemma.

**Lemma F.3** (Lemma 5 of (Wang et al., 2014)). *Suppose $p$ is a pdf with support in $[0, 1]$ and such that $p(x) \geq p_0 > 0$. Let $x_1, ... x_n$ be a sorted list of iid draws from $p$. Then, with probability at least $1 - 2p_0 n^{-10}$, the maximum gap between two draws satisfies*

$$\max_{i>1} |x_i - x_{i-1}| \leq \frac{c \log n}{p_0 n}$$

*where $c$ is a universal constant.*

By including the bad event of Lemma F.3 in a union bound, we have the following corollary of Theorem F.1. Notice that now, $TV_1[f; \mathcal{D}_X]$ is a random variable depending on the sampled covariates. In the special case where $f$ is differentiable, then $TV_1[f; \mathcal{D}_X] \leq \|f\|_{TV_1}$ a.s.

**Corollary F.4.** *Suppose the same setting as Theorem F.1, except that $x_1, ..., x_n$ are sorted draws from a pdf $p$ with support in $[0, 1]$ and such that $p(x) \geq p_0 > 0$. Then, with probability at least $1 - p_0 n^{-10} - \delta$, the error satisfies*

$$\sum_{t=1}^n (\hat{y}_t - f(x_t))^2 = \tilde{O}(n^{1/5} TV_1[f; \mathcal{D}_X]^{2/5})$$

*where $\tilde{O}$ hides constants (including $\sigma$) and polylog factors of $n$, $p_0$ and $\delta$.*

Similarly, for AKORN's Theorem F.2:

**Corollary F.5.** *Suppose now that $x_1, ..., x_n$ are sorted draws from a pdf $p$ with support in $[0, 1]$ and such that $p(x) \geq p_0 > 0$. Then, with probability at least $1 - p_0 n^{-10} - \delta$, the error of $\hat{f}$ satisfies*

$$\sum_{t=1}^{n}(\hat{f}(x_t) - f(x_t))^2 = \tilde{O}(n^{1/5} TV_1[f; \mathcal{D}_X]^{2/5})$$

*where $\tilde{O}$ hides constants (including $\sigma$) and polylog factors of $n$, $p_0$ and $\delta$.*

## G. Experimental Details and Additional Simulations

### G.1. Details

A single run of AKORN, Oracle Trend Filtering, and DoF Trend Filtering is performed as follows

1. Generate $\epsilon \sim \mathcal{N}(0, \sigma^2 I_n)$

2. Get $\hat{f} = \text{AKORN}(\{x_i, f(x_i) + \epsilon_i\}, \sigma)$

3. Get $\hat{f}_{tf}^{\lambda}$ for data $\{x_i, f(x_i) + \epsilon_i\}$ and parameter $\lambda$, We use the library `glmgen`: https://github.com/glmgen/glmgen.

4. Let $\hat{f}_{o-tf}$ be the $\hat{f}_{tf}^{\lambda}$ which has the smallest MSE with respect to the *noiseless data*

5. Let $\hat{f}_{s-tf} = \hat{f}_{tf}^{\lambda}$ where $\lambda = \arg\min_{\lambda \in E}\{\|\hat{f}_{tf}^{\lambda} - Y\|_2^2 + 2\sigma^2 L(\hat{f}_{tf}^{\lambda})\}$ where $L(g)$ gives the number of linear pieces of $g$.

6. Produce fitted values $\hat{y} = [\hat{f}(x_1)...\hat{f}(x_j)]$ and $\hat{y}_{o-tf} = [\hat{f}_{o-tf}(x_1)...\hat{f}_{o-tf}(x_j)]$ and $\hat{y}_{s-tf} = [\hat{f}_{s-tf}(x_1)...\hat{f}_{s-tf}(x_j)]$ for comparison

This procedure is used as a subroutine for producing Tables 1, 2, and Figure 2.

### G.2. ADDLE is worse than AKORN

In this appendix, we back up the statement made in Section 5.2 that ADDLE is not competitive with offline methods. In Figure 5, we reproduce a couple entries of Figure 2, substituting TF-DoF with ADDLE.

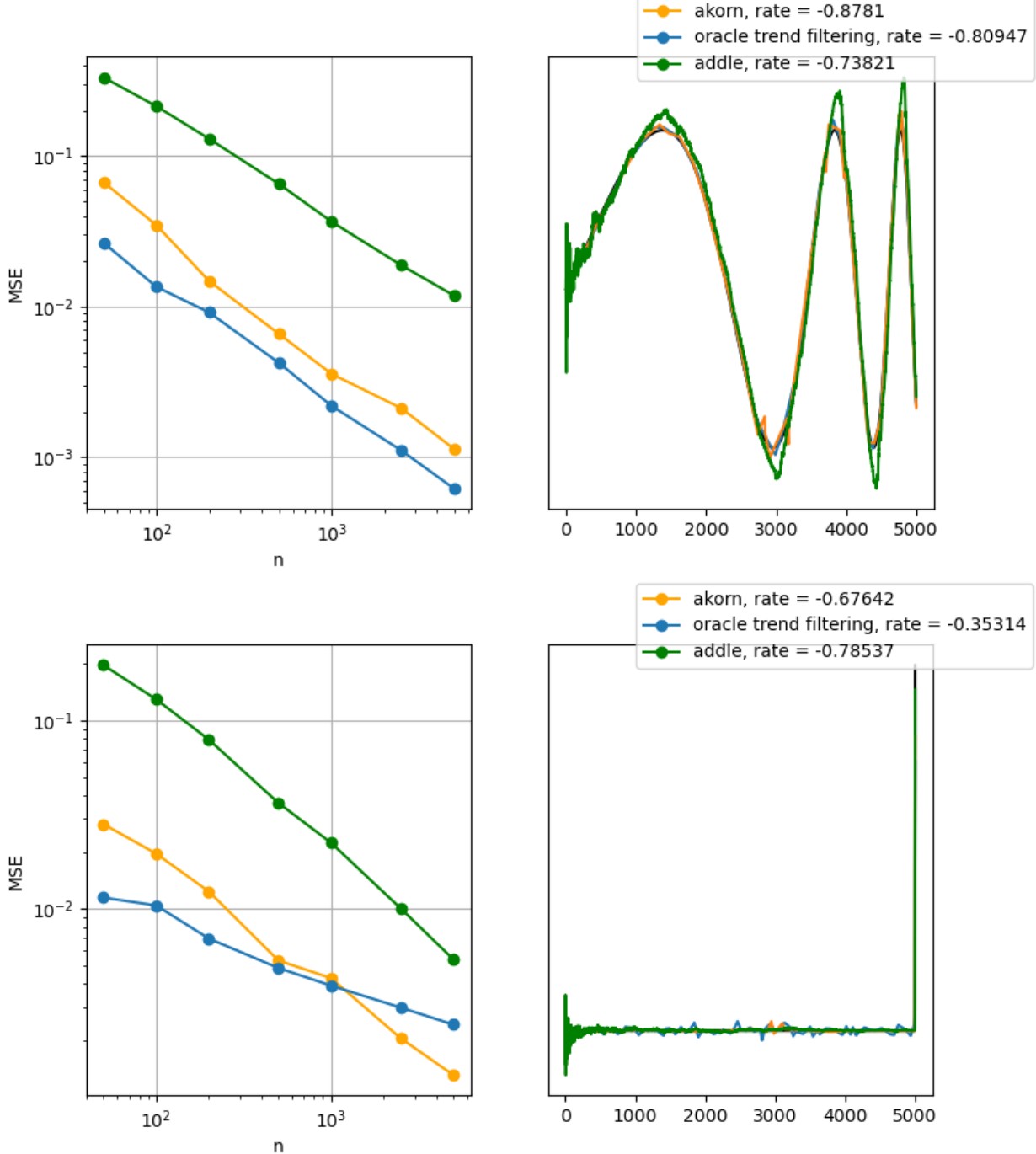

*Figure 5.* Same methodology as Figure 2 applied to the Doppler function and Jump Function, but this time comparing ADDLE to AKORN and Trend Filtering. There is roughly an order of magnitude difference in the MSEs for all $n$

