# OpenReview forum: "AKORN: Adaptive Knots generated Online for RegressioN splines"
_ICML.cc/2025/Conference — ICML 2025 poster_

### Official Review · Reviewer_V1hW · 2025-03-12

**Overall Recommendation:** 3

**Summary:**

This paper introduces AKORN (Adaptive Knots generated Online for RegressioN splines), a parameter-free algorithm for offline non-parametric regression over total variation (TV1)-bounded functions. AKORN leverages online learning techniques to automatically adapt knot selection for spline regression, eliminating the need for oracle knowledge of function smoothness, which is a major drawback of traditional non-parametric regression methods. The contributions of this paper include the following points
1. It achieves optimal rates without hyperparameter tuning (it adaptively selects knots based on change points in function smoothness).
2. It proposes a new online learning-based method for knot selection: AKORN uses ADDLE (Adaptive Denoising with Linear Experts), an online regression technique, to identify smooth and rough regions of the function.
3. Unlike existing online methods that output noisy pointwise predictions, AKORN reconstructs a continuous function.
4. Its Computational Efficiency and Scalability. It runs in O(n^2) time (comparable to other spline methods).

**Claims And Evidence:**

Yes

**Essential References Not Discussed:**

The paper has discussed a sufficient amount of relevant papers for showing and understanding their contributions.

**Experimental Designs Or Analyses:**

For the experiment, AKORN selects knots adaptively, but the paper does not test how sensitive its performance is to different knot placements, i.e., what happens if AKORN misidentifies some change points? This issue seems important to their proposed method.

**Methods And Evaluation Criteria:**

This paper introduces AKORN (Adaptive Knots generated Online for RegressioN splines). It makes sense to non-parametric regression problems.

**Other Comments Or Suggestions:**

Please see "Other Strengths And Weaknesses".

**Other Strengths And Weaknesses:**

Strength is clear: Their proposed method AKORN (Adaptive Knots generated Online for RegressioN splines) can automatically adapt knot selection for spline regression, eliminating the need for oracle knowledge of function smoothness, which is a major drawback of traditional non-parametric regression methods. Moreover, it achieves optimal rates without hyperparameter tuning (it adaptively selects knots based on change points in function smoothness).

Weakness:
1. The method assumes that covariates $x_i$ are equally spaced, which is a strong and unrealistic assumption in non-parametric regression. Although the paper mentioned that but it is a starting point for many non-parametric regression algorithms, it weakens the contribution of this paper significantly. For example, some other papers on trend filtering and wavelets have been extended to handle uneven spacing (Wang et al., 2014; Sadhanala & Tibshirani, 2019), but AKORN has not.
2. The paper only considers TV1-bounded functions, but many real-world applications involve smoother functions (e.g., TV2, TV3).
3. The paper claims AKORN achieves an instance-dependent rate Theorem 6.1-6.2, but there is no worst-case analysis. Trend filtering and spline methods have worst-case minimax bounds, ensuring robustness across all function classes.
4. The method is designed for one-dimensional regression problems. There is no discussion on whether AKORN extends to multivariate or high-dimensional settings.

**Questions For Authors:**

Please see the four questions of weakness in "Other Strengths And Weaknesses".

**Relation To Broader Scientific Literature:**

The key contributions are related to non-parametric regression, eliminating the need for oracle knowledge of function smoothness, which is a major drawback of traditional non-parametric regression methods. This is a commonly studied statistical and machine learning problem.

**Theoretical Claims:**

Yes, I checked the proof of their main results Theorem 6.1-6.2 and they look correct to me.

---

> ### Author Rebuttal · Authors · 2025-03-29
>
> Thank you very much for your consideration.
>
> ### Weaknesses 2 and 4
> The weaknesses you point out in 2 and 4 are fully accurate, although we feel that the multivariate problem you mention in 4 is outside of the scope of this paper. High-dimensional nonparametric regression is often considered separately from univariate regression. If you are curious about the challenges that remain in extending AKORN to $TV_{k > 1}$, we expand on them in our response to Reviewer YjPY.
>
> ### Worst-case analysis (Weakness 3)
> To be clear, AKORN is minimax optimal over $TV_1(C) = \\{f \ : \ TV_1(f) \leq C\\}$, which is the only minimax result we are aware of for linear Trend Filtering. This is a corollary of the instance-dependent results.
>
> ### Uneven Covariates (Weakness 1)
> With respect to the spacing of the covariates. Since submitting this paper, we have discovered that we can generalize AKORN to handle uneven covariates by tweaking our proof in a few places. Specifically, ADDLE/AKORN can achieve the same rate for covariates $x_1, … x_n \sim p$, where $p$ is some density that is bounded below (that is, $p(x) \geq p_0 > 0$ for $0 \leq x \leq 1$ exactly as in [4]). We will list the technical steps as a the end of this rebuttal. If reviewers and AC approve, we would like to include this result in the paper. In order to maintain accessibility and consistency, we believe it makes sense to include the result in an appendix, as is done in Tibshirani 2014.
>
> ### “What happens if AKORN misidentifies some change points?”
> While it would be an interesting experiment to see how many knots we could perturb while retaining good performance, do note that our theoretical guarantee says that AKORN will *not* catastrophically misidentify knots. Another lens on this point is as follows. Each time we run AKORN with different realizations of the noise vector, it can potentially identify a different knot-set. But our theoretical and experimental results show that whatever knot-set AKORN selects is sufficient for (near-)optimal performance (whp). As a practical observation, the sensitivity of the knot-set to noise increases with $n$, but the sensitivity of the MSE to the knot-set decreases with $n$.
>
> ## Technical steps for uneven covariates:
>
> In this response, we explain the tweaks necessary to prove that ADDLE/AKORN achieve the rate $\tilde O(n^{-4/5}C^{2/5})$ with probability $1 - 2p_0n^{-10} - \delta$  when covariates are sampled iid from a density $p$ such that $p(x) \geq p_0 > 0$ for $0 \leq x \leq 1$.
>
> **Key lemma, "Lemma K"** (The bias of  linear regression error is controlled even on uneven points)
> Let $x_1, ..., x_n$ be a sorted list of design points whose max gap is bounded: $\max_{i > 1\} x_i - x_{i - 1} = O(\log{n}/n)$.
> Let $l_{r:s}$ be the linear least squares fit on $(x_r, \theta_r), ... (x_s, \theta_s)$. Then $\sum_{j = r}^s (l_{r:s}(x_j) - \theta_j)^2 = \tilde O(\frac{|s - r|^3TV_1(\theta[r:s])^2}{n^2})$.
>
>
> **Lemma 5 from [4]**
> Suppose $\{x_i\}$ is sampled iid from a pdf, p, that is bounded below by $p_0 > 0$. Lemma 5 from [4] implies that, w.h.p, the max gap between any two covariates in the set $\{x_i\}$ is bounded by $O(log(n)/n)$ with probability at least $1 - 2p_0n^{-10}$.
>
> **ADDLE generalization:**
> 1) Change online linear regression experts to (clipped) VAW forecasters. This very minor algorithmic change does not affect computational efficiency.
> 2) **VAW linear forecasters enjoy the same upper bound as offline linear regression up to an additive constant**
> This follows from Theorem 11.8 in [1] paired with Corollary 40 in [2] to control the norm of the least-squares comparator.
> 3) We can reduce the error of ADDLE on an interval $[r, s]$ to the error of the (clipped) expert that starts at $r$ using the same argument as before (i.e., Appendix D's proof up to Equation (15) doesn’t need to change).
> 4) The error (unclipped) expert  can then be bounded using part (2). Instead of line 1157 of the submission, we use part (2) above to bound the online experts error by  $\sum_{j = r}^s (\hat l_{r:s}(x_j) - \theta_j)^2 + O(1)$, where $\hat l_{r:s}$ is batch regression of the noisy responses $y_r, ... y_s$ onto $x_r, ... x_s$. We then use concentration of $\hat l_{r:s} $ to $E[\hat{l}_{r:s}]$ and Lemma "K" to finish the bound.
> 5) We can use the same oracle partitioning scheme as before to produce a set of intervals of size $O(n^{1/5}C^{2/5})$ together with experts who achieve constant error on each interval.
>
> **AKORN generalization**
> The proof for AKORN now needs only trivial changes.
>
> More detailed treatment can be found at this anonymized link: https://anonymous.4open.science/r/AKORN-uneven-D6FB/akorn_non_evenly_spaced_points.pdf
>
> [1] Prediction Learning and Games. Cesa-Bianchi and Lugosi.
>
> [2] Adaptive Online Estimation of Piecewise Polynomial Trends. Baby and Wang, 2020
>
> [3] Multivariate trend filtering for lattice data, Sandhalana et al., 2024.
>
> [4] The falling factorial basis and its statistical applications. Wang et al., 2014

---

> > ### Comment · Reviewer_V1hW · 2025-04-04
> >
> > I thank the authors for all the detailed response for my questions. While I acknowledge this work's novelty and find it interesting, as the authors' response suggests, it might still not be a quite complete work or at least there are many things that could have been added to the paper before publishing in a top conference such as ICML.

---

> > > ### Author Response · Authors · 2025-04-04
> > >
> > > Thanks for the follow-up discussion.
> > >
> > > We hope your main concern on the "worst-case optimality" was addressed.
> > >
> > > As we responded, the uneven covariate case is straightforward and adds no new technical challenge.  We plan to add it to the appendix of the paper so as to keep the notations clean in the main paper.
> > >
> > > For higher dimensional and higher order TV, they are non-trivial generalizations and we believe it is better for the current paper to focus on 1D.  Notice that this is the first result of its kind that converts the selected knots of an online algorithm to a valid nonparametric regression fit. We believe it is better to focus on explaining the idea and results clearly than stacking more theorems and results in generalized settings.
> > >
> > > Thank you again!

---

### Official Review · Reviewer_YjPY · 2025-03-12

**Overall Recommendation:** 3

**Summary:**

This paper proposes AKORN, a novel approach for offline non-parametric regression that adaptively selects spline knots without requiring manual hyperparameter tuning. The proposed method yields estimators competitive with oracle-enhanced Trend Filtering, attaining near-optimal theoretical performance for TV-bounded regression functions. The theoretical guarantees provided are thorough and rigorously developed, showcasing that AKORN's performance is competitive with the state-of-the-art offline methods.

**Claims And Evidence:**

I think the claims are clearly supported.

**Essential References Not Discussed:**

None

**Experimental Designs Or Analyses:**

Experimental design is generally sound. The synthetic functions (piecewise linear, Doppler, Jump) ensure theoretical and empirical coherence. However, there is no performance stability discussion

**Methods And Evaluation Criteria:**

The methods and evaluation criteria, such as Doppler, jump functions, and evaluation metric (mean squared error, MSE), are appropriate.

**Other Comments Or Suggestions:**

I think the structure of the article needs to be readjusted. Articles usually start with theoretical analysis and then conduct experimental verification. Your appendix is very rich and can fill eight pages.

**Other Strengths And Weaknesses:**

Strengths:
1. The proposed AKORN algorithm is highly original in combining offline spline regression with adaptive online learning.
2. The theoretical framework and results are well-established.

Weaknesses:
1. In the Proof Sketches section, the statements of theorems are all informal.
2. Practical applications on real-world datasets would greatly strengthen the paper’s appeal to applied researchers and practitioners.
3. The assumption of equally spaced covariates is restrictive. The authors acknowledge this limitation but should ideally outline a more explicit path forward for generalizing to unevenly spaced data.
4. What are the primary theoretical or computational barriers to extending AKORN to higher-order smoothness classes?

**Questions For Authors:**

See Weaknesses.

**Relation To Broader Scientific Literature:**

AKORN builds upon previous work on trend filtering, wavelet smoothing, and online regression techniques. Unlike prior methods, AKORN adaptively chooses knots without needing explicit smoothness information. It extends influential approaches such as Trend Filtering (Kim et al., 2009; Tibshirani, 2014) and wavelet-based smoothing (Donoho & Johnstone, 1998), providing an integrated, parameter-free alternative that automatically adjusts to the underlying structure of data.

**Theoretical Claims:**

I checked Theorem 6.1 (Bound on ADDLE error) and Theorem 6.2 (Bound on AKORN MSE). During the proof, the "Change-point Detection Lemma" (Lemma 7.2) is pivotal, rigorously connecting the adaptively chosen knots to a sparse and statistically efficient linear spline. However, the Lemma 7.2 is informal and the authors didn't provide the formal expression or the source. Additionally, in the Proof Sketches section, the statements of theorems are all informal. Why doesn't the author provide a formal statement directly?

---

> ### Author Rebuttal · Authors · 2025-03-29
>
> Thank you for your attention to this work!
>
> ### Non-evenly spaced design points
> Since submitting this paper, we have discovered that we can generalize AKORN to handle uneven covariates by tweaking our proof in a few places. Specifically, ADDLE/AKORN can achieve the same rates for covariates $x_1, … x_n \sim p$ where $p$ is a pdf that is bounded below (as in Wang et al., 2014). We enumerate the necessary changes in our response to Reviewer V1hW and would like to include the generalized result as an appendix of the final version of this paper (as is done in Tibshirani, 2014). Your input on this extension would be greatly appreciated.
>
> ### Proof sketches
> Thank you for mentioning the informality in the proof sketches section. The Change-point Detection lemma that you mention is an informal statement of Lemma C.1 in the Appendix, which we forgot to mention during the proof sketch. This will be fixed in future versions of the paper. More generally, the reason that the proof-sketch lemmas are informally stated is space. We hope that the informal statements are understandable, and welcome input on this front (we are aware of a typo in Lemma 7.3 – the projection operator should be applied to Y rather than $\theta$).  Rigorous statements are all available in the appendices, and the main theorems of the paper are formally stated in Section 6.
>
> ### Higher-order Smoothness Classes
> There are two main steps to generalizing to higher-order smoothness classes. The first (relatively easy) step is generalizing the computations in Appendix E.1 to polynomial regression. For unevenly spaced design points, this corresponds to the generalizing the (much cleaner) "Lemma K" given in the response to V1hW. The second task is in generalizing the Spline Existence Lemma (Lemma C.7) to higher order splines. Specifically, given two kth-order piecewise polynomial functions with disjoint knot-sets, we need to show that there is a kth-order spline on some augmented knot-set that lies between the two curves (or some strategic weakening of this).  We do not yet know how to do this.  Apart from these steps, the bulk of the proof can be adapted to the higher-order cases fairly trivially.

---

### Official Review · Reviewer_Rwji · 2025-03-12

**Overall Recommendation:** 3

**Summary:**

This paper studies the non-parametric regression over TV_1-bounded functions. The paper proposes a parameter-free algorithm (AKORN) which leverages online learning techniques to select knots for regression splines. The algorithm proposed achieves near-optimal rates without hyperparameter tuning. Both theoretical and empirical results are presented.

**Claims And Evidence:**

As a theoretical work, most of the claims are well-supported by providing rigorous proof. Given that most related work on this problem achieves O(n) or O(nlogn) computational complexity, it would better to explore in detail how to reduce the O(n^2) complexity in this paper with the geometric cover trick.

**Essential References Not Discussed:**

I am not aware of essential references that are missing.

**Experimental Designs Or Analyses:**

The experimental design is sound.

**Methods And Evaluation Criteria:**

The proposed methods and evaluation criteria make sense for the problem studied under the setting of this work.

**Other Comments Or Suggestions:**

Few typos:
1.\hat{y_i} should be replaced with \hat{y}_i on line 046
2."where \tilde{O} and \lesssim hide..." should be replaced with "where \lesssim hides..." on line 698
3. Using T to denote transposing is a bit misleading (for example on line 170)

**Other Strengths And Weaknesses:**

Strengths:
-The idea of leveraging online learning techniques and learning forward and backwardly to design the adaptive method is interesting.
-Processes data sequentially, adjusting knots based on residuals, similar to no-regret learning techniques.
-AKORN does not require tuning smoothing penalties or pre-specifying the number of knots.
-Proposed method matches minimax-optimal convergence rates for TV₁ functions.
-The experimental results demonstrate that AKORN performs competitively with oracle-tuned Trend Filtering.

Weaknesses:
-Uniform spacing avoids knowing smoothness knowledge but uniform spacing is also a strong assumption as it reveals more structural information as knowing smoothness.
-As a consequence, it cannot directly handle scattered data or missing time-series values without preprocessing.
-Given the existence of several methods achieves a minimax-optimal convergence rate and comparable runtime, it's unclear if AKORN provides significant advantages over other adaptive regression methods.

**Questions For Authors:**

I have no further questions for the authors.

**Relation To Broader Scientific Literature:**

n/a

**Theoretical Claims:**

Reviewed the proof sketches in the main body and verified the theorem and lemma statements, along with select proof details in the appendix. The overall proof structure appears sound and well-reasoned.

---

> ### Author Rebuttal · Authors · 2025-03-29
>
> Thank you very much for this detailed review. Firstly, thank you for mentioning the typos. We will make appropriate adjustments (e.g. remove the use of $T:=n$ to avoid confusion with the transpose operation).
>
> ### Uniform spacing:
> Since submitting this paper, we have discovered that we can generalize AKORN to handle uneven covariates by tweaking our proof in a few places. Specifically, we can achieve the same result for covariates $x_1, … x_n \sim p$ and $p$ bounded below (exactly as in Wang et al., 2014). In light of reviewers’ comments, we would like to include this result as an appendix. In our response to Reviewer V1hW, we outline the tweaks that are required. These turn out not to be too extensive; from a technical perspective, fixed uniform design is not too different from iid samples from a (nice) distribution. We would greatly appreciate your feedback on this point.
>
> ### “Given the existence of several methods achieves a minimax-optimal convergence rate and comparable runtime, it's unclear if AKORN provides significant advantages over other adaptive regression methods.”
>
> As you mention, AKORN is adaptive and empirically competitive with Trend Filtering, Locally Adaptive Regression Splines. We aren’t aware of other adaptive methods with comparable empirical performance (see comparison with Wavelets in Section 5 of the paper).
>
> ### Computational complexity:
>
> It’s true that the computational complexity of AKORN is slightly prohibitive, but we elaborate on the topic now.
>
> Firstly, note that the $O(n^2)$ upper bound is, in practice, somewhat loose. This is because AKORN periodically restarts ADDLE during the online pass, meaning that we do not perform $t$ regression steps at timestep $t$. Furthermore, when we perform regression onto the knotset K, whose size is $d = O(n^{1/5}C^{2/5})$, we use $O(d^2n) = O(n^{7/5}C^{2/5})$ compute. So even without any tweaks, real performance is often better than the $O(n^2)$ suggests.
>
> On top of this, it is possible (as you mention) to run ADDLE with a smaller set of experts to get an $O(n\log{n})$ online algorithm. In practice, this may lead to significant speed-up when ADDLE is called as a subroutine by AKORN. However, due to the definition of AKORN, this doesn’t shave any compute off of the asymptotic runtime on the offline problem (at least not trivially).
>
> Lastly, allow us to note that, technically speaking, the worst-case compute of SOTA Trend Filtering (TF) is superlinear: $O(n^{3/2}\log{1/\epsilon})$ to output an $\epsilon$-suboptimal solution for the associated optimization problem (Tibshirani, 2014). On the other hand, AKORN computes an exact solution.
>
> Furthermore, the $O(n^{3/2}\log{1/\epsilon})$ bound for Trend Filtering doesn’t take into account the cost of parameter tuning. If we do this with SURE, for an upper bound $C$ on $TV_1[f]$ and at discretization level $\Delta$, then we need to solve TF $C/\Delta$ times while also computing DoF for each solution. Furthermore, in general, the only a-priori upper bound on $C$ that is possible is $C = O(n^2)$.
>
> So while, practically speaking, people know how to do TF extremely efficiently, the theoretical picture is slightly complicated.

---

> > ### Comment · Reviewer_Rwji · 2025-04-06
> >
> > Thank you for your responses, particularly the part regarding computational complexity. I have carefully considered your rebuttal, along with the comments from the other reviewers, and I have decided to maintain my evaluation as a borderline acceptance.

---

### Official Review · Reviewer_tkPX · 2025-03-14

**Overall Recommendation:** 3

**Summary:**

The authors consider the problem of nonparametric regression over the class of $TV_1$-bounded functions. Crucially, the authors aim to overcome the issue of needing oracle knowledge regarding certain features of the data-generating process, while still achieving optimal error rates. Despite being in an offline setting, the authors leverage existing online denoising algorithm with both forward and backward passes over the offline data. The authors show that their approach performs empirically well relative to other baselines in the literature.

**Claims And Evidence:**

Yes, the claims seem reasonable given the empirical performance of this approach.

**Essential References Not Discussed:**

I am unaware of other essential references not mentioned here.

**Experimental Designs Or Analyses:**

N/A

**Methods And Evaluation Criteria:**

All experiments are reasonable evaluations of their method to baselines that (1) take oracle knowledge, which serves as an upper bound on  performance (i.e., impressive to perform as good as this) and (2) same theoretical guarantees without oracle knowledge, which serves as the baseline to beat. One minor change that would be helpful in assessing numerical results would be to have error bars on Tables 1 and 2.

**Other Comments Or Suggestions:**

N/A

**Other Strengths And Weaknesses:**

N/A

**Questions For Authors:**

N/A

**Relation To Broader Scientific Literature:**

To the best of my knowledge, this would be the second work that provides optimal error rates beyond another baseline that these authors test. However, this work improves upon the empirical performance of the first paper greatly.

**Theoretical Claims:**

N/A

---

> ### Author Rebuttal · Authors · 2025-03-29
>
> We appreciate your time and consideration.

---

### Decision · Program_Chairs · 2025-05-01

**Decision:**

Accept (poster)

**Comment:**

This paper presents a parameter-free algorithm for offline non-parametric regression over TV-1 bounded functions. The proposed method (referred to as AKORN) does not require knowledge of the ground-truth smoothness, and results in an estimator whose empirical performance is similar to an oracle Trend Filtering baseline.

The reviewers unanimously agree that the paper is well-written, and its contributions merit acceptance. After carefully reading the rebuttal/discussion, I tend to agree. Please incorporate the following comments in the final version of the paper. In particular, addressing the following concerns will help strengthen the current version of the paper:
- Including the proof handling uneven covariates (response to Rev. Rwji, Rev. YjPY)
- Clear description of the method's computational complexity and its comparison with existing work  (response to Rev. Rwji)
- Including the discussion about higher-order smoothness classes (response to Rev. YjPY, Rev. V1hW)